# Evolutionary conservation of the grape sex-determining region in angiosperms and emergence of dioecy in Vitaceae

Mélanie Massonnet [1] ✉, Noé Cochetel [1], Valentina Ricciardi [2,3], Andrea Minio [1], Rosa Figueroa-Balderas[1], Jason P. Londo[4] & Dario Cantu [1,5] ✉

In bunch grapes (*Vitis* spp.), flower sex is controlled by a ~200-kilobase sex-determining region (SDR) that contains genes involved in floral development. Here, we show that this region evolved from an ancient, highly conserved locus across angiosperms. Comparative genomic analysis of 56 plant genomes identifies homologous regions in all flowering plants but not in non-flowering lineages, suggesting a conserved role in floral function. Within the grape family (Vitaceae), we assemble and phase SDR haplotypes from six species, plus *Leea coccinea* as an outgroup, and find strong structural conservation, with size variation largely attributable to repetitive elements. Among the dioecious genera, *Vitis* and *Muscadinia* exhibit suppressed recombination in the SDR and share candidate sex-determining genes, whereas in *Tetrastigma*, the region appears to remain recombining, pointing to an alternative mechanism of sex determination. Altogether, our results suggest that dioecy emerged in grapes from a deeply conserved, collinear genomic region composed of multiple genes involved in floral development, morphology, and sexual fertility.

Dioecy, the presence of distinct male and female individuals within a species, is rare, occurring in approximately only 6% of flowering plants[1]. Dioecy has evolved independently multiple times from hermaphroditic ancestors in angiosperms, with estimates suggesting between 900 and 5000 transitions across 175 plant families[1]. Interestingly, dioecy often reverts back to hermaphroditism, underscoring the remarkable plasticity and adaptability of plant reproductive strategies[1,2].

In *Vitis* spp., or bunch grapes, all ~80 species are dioecious, except for the domesticated grapevine (*Vitis vinifera* ssp. *vinifera*), which produces hermaphroditic flowers[3]. In *Vitis* spp., flower sex is determined by a sex-determining region (SDR) located on chromosome 2 and spanning approximately 200 kbp[4,5]. The *Vitis* SDR exhibits a high linkage disequilibrium with boundaries strictly conserved across the genus[4,5]. Divergence time estimates suggest that dioecy in *Vitis* originated at least ~20 million years ago (Mya)[5]. The *Vitis* SDR is composed of twelve to fourteen protein-coding genes[5], with five genes involved in floral organ development and identity in plants (Table 1). These include the two candidate sex-determining genes *VviYABBY3* and *VviINP1*[4,5], and *VviPLATZ1*, which has been shown to influence stamen morphology[6]. The multiple closely linked flowering-related genes within the SDR raise questions about whether this region is unique to the *Vitis* genus or if it is also present in the broader Vitaceae family, or even in more distantly related plants.

The *Vitis* genus belongs to the Viteae clade within the Vitaceae family (Fig. 1)[7,8]. The Vitaceae family is divided into five major clades: (i) Ampelopsideae, (ii) Parthenocisseae, (iii) Viteae, (iv) Cayratieae, and (v) Cisseae[9]. The family comprises 16 genera and ~950 species

[1]Department of Viticulture and Enology, University of California Davis, Davis, CA, USA. [2]Dipartimento di Scienze Agrarie e Ambientali, Università degli Studi di Milano, Milan, Italy. [3]National Research Council (CNR), Institute of Biosciences and BioResources (IBBR), Division of Palermo, Palermo, Italy. [4]School of Integrative Plant Science, Cornell University, Geneva, NY, USA. [5]Genome Center, University of California Davis, Davis, CA, USA. ✉e-mail: mmassonnet@ucdavis.edu; dacantu@ucdavis.edu

**Table 1 | Homologs of the *Vitis* SDR genes have a role in flower development and sex fertility**

| Functional description | Plant species | Gene name | Mutant | Phenotype | Reference |
|---|---|---|---|---|---|
| YABBY transcription factor | *Arabidopsis thaliana* | *AtYABBY1, AtFIL, AtAFO* | Knockout *fil* | Altered floral organ number, position, identity, and development during flower development. Partial male sterility. | 97 |
| | *Oryza sativa* | *OsYABBY4* | CAM35S-*OsYABBY4* | Increased number of stamens, pistils, and stigmas. | 98 |
| SKU5 | *Arabidopsis thaliana* | *AtSKU5* | Knockouts *sks1,sks2,sks3,sku5* | Smaller flowers than wild type. Reduced seed production due to maternal-dependent ovule abortions. | 99 |
| Vacuolar invertase | *Arabidopsis thaliana* | *AtBETAFRUCT4* | Knockouts *Atvi2-1, Atvi2-2* | Reduced pollen germination. | 100 |
| INAPERTURATE POLLEN1 | *Arabidopsis thaliana* | *AtINP1* | Knockout *atinp1* | Complete lack of pollen aperture. | 101 |
| | *Oryza sativa* | *OsINP1* | Knockout *osinp1* | Complete lack of pollen aperture. Sterile pollen. | 102 |
| | *Zea mays* | *ZmINP1* | Knockout *zminp1* | Complete lack of pollen aperture. Sterile pollen. | 103 |
| | *Eschscholzia californica* | *EcINP1* | Silencing pTRV2:*EcINP1* | Abnormal apertures. | 104 |
| PLATZ transcription factor | *Vitis vinifera* | *VviPLATZ1* | Knockout *vviplatz1* | Reflex filaments. | 6 |

distributed across tropical and temperate regions, characterized by their distinctive leaf-opposed tendrils[9]. Vitaceae exhibit significant morphological diversity, particularly in vegetative structures, seed characteristics, inflorescence types, and floral features[10,11]. There is also considerable variation in chromosome numbers, ploidy levels, and genome sizes both among and within Vitaceae clades[12,13]. For example, *Vitis* ssp. are diploid with 19 chromosomes (2n = 38), individuals of the second grape genus (*Muscadinia* spp.) are diploid with 40 chromosomes (2n = 40), and some *Cissus* species exhibit tetraploidy or hexaploidy[13]. While most Vitaceae species are hermaphroditic, other mating systems, including dioecy, also occur. Dioecy is found within the Viteae clade, which includes the grape genera *Vitis* and *Muscadinia*, as well as the *Tetrastigma* genus of the Cayratieae clade[9].

*Vitis* and its sister genus *Muscadinia*[14–16] diverged approximately 18–47 Mya[17–19] and each consist of dioecious species[20]. A high-density linkage map has located the SDR in *Muscadinia rotundifolia* on the same chromosomal region as in *Vitis* spp.[21], suggesting that the SDRs of both genera have evolved from a common dioecious ancestor. However, the precise boundaries of the muscadine SDR have yet to be defined, making it unclear whether the same genes regulate flower sex in both grape genera. A previous study identified the same candidate male-sterility mutation in the female-associated (F) haplotype of the male *M. rotundifolia* cv. Trayshed as in *Vitis* species[4]. However, for the candidate female-suppressing gene in *Vitis* spp., *VviYABBY3*, Trayshed lacked the two non-synonymous SNPs specific to the male-associated (M) allele in *Vitis*, and the gene expression of *VviYABBY3* in muscadine flowers has yet to be investigated.

In this study, we explore the evolutionary history of the *Vitis* SDR over approximately 200 million years within plants and the Vitaceae family. To achieve this, we examine gene collinearity and conservation of the *Vitis* SDR across 56 plant genomes, encompassing a broad range of angiosperms from monocots to eudicots, as well as non-flowering plants. We also analyze the conservation of the *Vitis* SDR among four other Vitaceae clades by sequencing and assembling genomes of six Vitaceae species and the outgroup *Leea coccinea*. Furthermore, we assess whether the orthologous SDR region in *Vitis* can be associated with sex determination in the dioecious *Tetrastigma* species. Lastly, we investigate the muscadine grape SDR by defining its boundaries and identifying its candidate sex-determining genes. This comprehensive analysis sheds light on the evolution and functional significance of the *Vitis* SDR across diverse plant taxa.

## Results

### The *Vitis* sex-determining region is highly conserved in flowering plants

To investigate the evolutionary history of the *Vitis* SDR in plants, we analyzed the collinearity between the grapevine SDR and its homologous regions across 56 plant genomes (Supplementary Data 1), comparing these results to overall genomic collinearity. The dataset comprised 43 angiosperms (34 eudicots, one species of Ceratophyllales, four monocots, one species of magnoliids, one species of Chloranthales, two species from the ANA grade), and 13 non-flowering species: the gymnosperm *Taxus chinensis* (Chinese yew), the ancient tracheophyte *Selaginella moellendorffii* (spike moss), the bryophyte *Physcomitrium patens* (spreading earthmoss), and ten hornworts.

Gene collinear blocks between the haplotype 1 of *V. vinifera* ssp. *vinifera* cv. Cabernet Sauvignon (hereafter referred to as Cabernet Sauvignon) and the 63 plant haplotypes were detected using MCScanX[22]. Because the *Vitis* SDR is composed of twelve protein-coding genes, the collinear gene blocks were then compared with windows of twelve consecutive genes (2307 windows in total). Twelve-gene windows with the greatest number of collinear genes were selected as orthologous windows. In angiosperms, we identified an average of $1{,}958.4 \pm 218.0$ orthologous 12-gene windows, whereas non-flowering species showed significantly fewer windows ($94.6 \pm 45.0$). Importantly, an orthologous window corresponding to the *Vitis* SDR was detected in all angiosperms whereas it was absent in all thirteen non-flowering species (Supplementary Fig. 1). Among angiosperms, the number of collinear genes within the orthologous window of the *Vitis* SDR was significantly higher than those in orthologous windows of the other 19 grape chromosomes (Kruskal–Wallis test; $P$ value = $1.1 \times 10^{-15}$; Fig. 2a). Most genes of the *Vitis* SDR were conserved across angiosperm genomes, including both eudicots and monocots (Fig. 2e). Manual refinement of gene models identified thirteen additional protein-coding genes and eight pseudogenes within the *Vitis* SDR-orthologous window across eleven genomes (Supplementary Fig. 1 and Supplementary Data 2). Notably, the two candidate sex-determining genes in grapes, *VviYABBY3* and *VviINP1*, were located within the same orthologous window in 26 eudicots (76.5%) and the monocot *Phoenix dactylifera* (date palm; 25% of the monocots; Supplementary Fig. 1). Additionally, the rank-normalized gene conservation score of the collinear genes within the *Vitis* SDR-orthologous window was significantly greater than that of other collinear genes

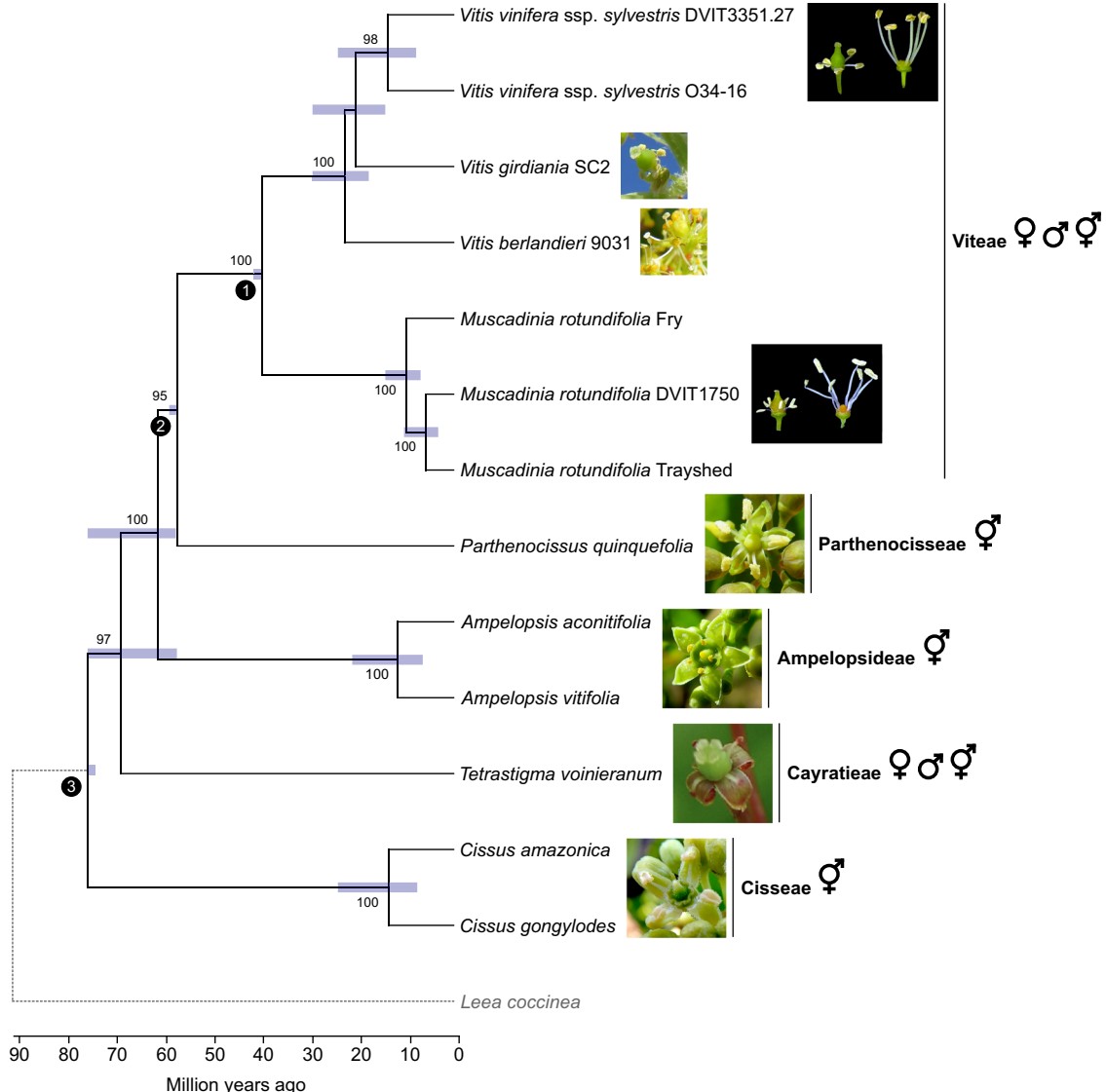

**Fig. 1 | Phylogeny of the Vitaceae and morphology of their flowers.** Phylogenetic tree predicted from single-copy orthologues in angiosperms. The branches represent divergence times in million years. Bars around each node represent 95% confidence intervals. *L. coccinea* was used as an outgroup to the Vitaceae family. Calibration points are depicted at the nodes (1–3) using black circles. Bootstrap values greater than 80 are indicated. The symbols ♀, ♂ and ⚥ represent the female, male, and hermaphrodite flower sex type present in each clade, respectively. Picture of a female flower from *Tetrastigma* was modified from ref. 105. Source data are provided as a Source Data file.

(Kruskal–Wallis test; *P* value = $1.7 \times 10^{-16}$; Fig. 2b). These findings collectively indicate that the genomic region corresponding to the *Vitis* SDR is highly conserved among angiosperms but absent in non-flowering species.

Because several plant species used in the gene collinearity analysis had undergone whole-genome duplication(s)[23], which could potentially affect the analysis of conservation of the *Vitis* SDR, we investigated whether gene collinearity was maintained in all homologous (i.e., orthologous and paralogous) regions by repeating the analysis using GENESPACE[24]. No synteny was detected between the SDR of haplotype 1 of Cabernet Sauvignon and the genome of the thirteen non-flowering plants. In contrast, among angiosperms, we identified at least one homologous region for an average of 1755.9 ± 347.5 12-gene windows. The number of homologous regions per collinear 12-gene window ranged from approximately 1.1 in *Cannabis sativa* and in each haplotype of *Fragaria* species, to 3.0 ± 1.1 in *Actinidia chinensis* and *Glycine max* (Supplementary Fig. 2). Indeed, four homologous regions corresponding to the Vitis SDR were found in *Actinidia chinensis* and *Glycine*

*max*, but also in *Arabidopsis thaliana* and *Daucus carota* ssp. *sativus* (Supplementary Fig. 3). All these plant species had experienced two whole-genome duplications since the ancient hexaploidization event γ[23,25,26]. In terms of number of collinear genes and their rank-normalized gene conservation score per 12-gene window, similar results were found when including both orthologous and paralogous regions, although the difference between the *Vitis* SDR and the other 19 grape chromosome was slightly reduced, but still significant (Kruskal–Wallis test; *P* value < $1 \times 10^{-9}$; Fig. 2c, d). These results support that the genomic region corresponding to the *Vitis* SDR is highly conserved among angiosperms

## The *Vitis* sex-determining region is conserved across the Vitaceae family

To investigate whether the genomic region corresponding to the *Vitis* SDR is conserved across the Vitaceae, we sequenced, assembled, and phased the genomes of six Vitaceae accessions from four major clades: *Parthenocissus quinquefolia* (Parthenocisseae clade), *Ampelopsis*

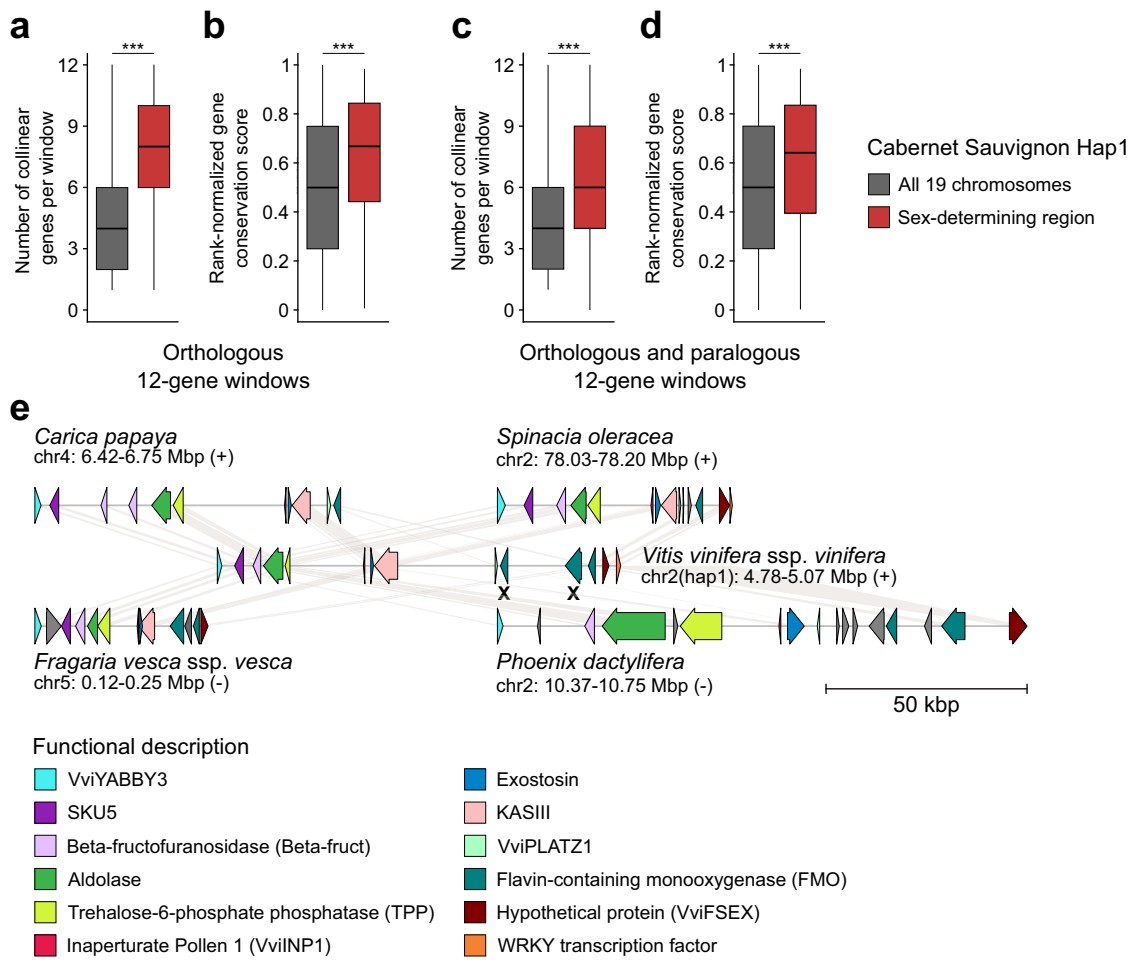

**Fig. 2 | Conservation analysis of the *Vitis* sex-determining region (SDR) genes in angiosperms.** Number of collinear genes per 12-gene orthologous (**a**) and orthologous and paralogous (**c**) windows in the nineteen chromosomes of Cabernet Sauvignon Haplotype 1 (Hap1) and the *Vitis* SDR among the angiosperm genomes ($n = 50$ haplotypes (43 angiosperms)) (Kruskal–Wallis test; **a** $P$ value = $1.1 \times 10^{-15}$; **c** $P$ value = $4.1 \times 10^{-10}$). Gene conservation score of the collinear genes within 12-gene orthologous (**b**) and orthologous and paralogous (**d**) windows in the Cabernet Sauvignon Haplotype 1 and the *Vitis* SDR among the angiosperm genomes ($n = 50$ haplotypes (43 angiosperms)) (Kruskal–Wallis test; **b** $P$ value = $1.7 \times 10^{-16}$; **d** $P$ value = $1.3 \times 10^{-13}$). **a–d** share the same color legend. The middle bars represent the median, while the bottom and top of each box represent the 25th and 75th percentiles, respectively, and the whiskers extend to 1.5 times the interquartile range. ***, $P$ value < $1 \times 10^{-9}$. **e** Schematic representations of the gene content in the orthologous regions of the *Vitis* SDR in four angiosperms. Each haplotype is represented with arrows to depict annotated genes. Pseudogenes are indicated with an X. The scale below the haplotypes denotes the length of the region. Source data are provided as a Source Data file.

*aconitifolia* and *A. vitifolia* (Ampelopsideae clade), *C. amazonica* and *C. gongylodes* (Cisseae clade), and *T. voinieranum* (Cayratieae clade), as well as *L. coccinea* as outgroup (Fig. 1 and Supplementary Data 3–5). The genome assembly sizes for *A. vitifolia* and *C. amazonica* were 785.6 Mbp and 704.3 Mbp, respectively, while *A. aconitifolia*, *P. quinquefolia*, and *C. gongylodes* had larger genome sizes of 999.1, 1136.4, and 1270.6 Mbp, respectively (Supplementary Data 5). The diploid genome of *T. voinieranum* was the largest of the study, with a size of 4419.2 Mbp. Regarding the outgroup *L. coccinea*, its diploid genome was 1104.9 Mbp long.

By aligning the SDR genes from Cabernet Sauvignon, we identified 21 regions homologous to the *Vitis* SDR: two in the outgroup *L. coccinea*, two in *P. quinquefolia*, two in *A. aconitifolia*, one in *A. vitifolia*, four in *C. amazonica*, eight in *C. gongylodes*, two in *T. voinieranum* (Fig. 3). The number of haplotypes found in *C. amazonica* and *C. gongylodes* suggest that these two species are tetraploid and octoploid, respectively. In *A. vitifolia*, only one *Vitis* SDR-homologous region (VSR) was identified. Alignment of short DNA sequencing reads from this accession revealed 72 heterozygous SNPs within the region (i.e., 0.36 SNPs per kbp), indicating a low heterozygosity, which may explain why only one haplotype was generated by FALCON-Unzip.

The gene content across the identified haplotypes was largely conserved (Fig. 3). We could observe some differences between clades and even species within clades. For example, in *A. aconitifolia*, we identified seven and five genes encoding flavin-containing monooxygenases (FMOs) in the first and second haplotypes, respectively, while the region in *A. vitifolia* had four FMO genes in its single haplotype. The number of FMO genes also varied between the two *Cissus* species, and the gene encoding the hypothetical protein *VviFSEX* was absent in all *Cissus* haplotypes. Additionally, the *VviINP1* gene was present in half of the haplotypes in both *C. amazonica* and *C. gongylodes*. We further investigated whether the deleterious 8-bp deletion in the candidate sex-determining gene *VviINP1* was present in other Vitaceae. Alignment of DNA-seq short reads from 159 Vitaceae accessions, representing 88 species, revealed that this mutation was only found in the *Vitis* genus and *M. rotundifolia* (Supplementary Data 6).

Despite the high conservation of the gene content, the size of the VSRs varied significantly between clades, species, and haplotypes. The region from *VviYABBY3* to the WRKY TF gene was the shortest in the two *Cissus* species ($86.5 \pm 4.9$ kbp), followed by *P. quinquefolia* (144.8 and 147.2 kbp), the two *Ampelopsis* species ($193.7 \pm 8.7$ kbp), and *T.*

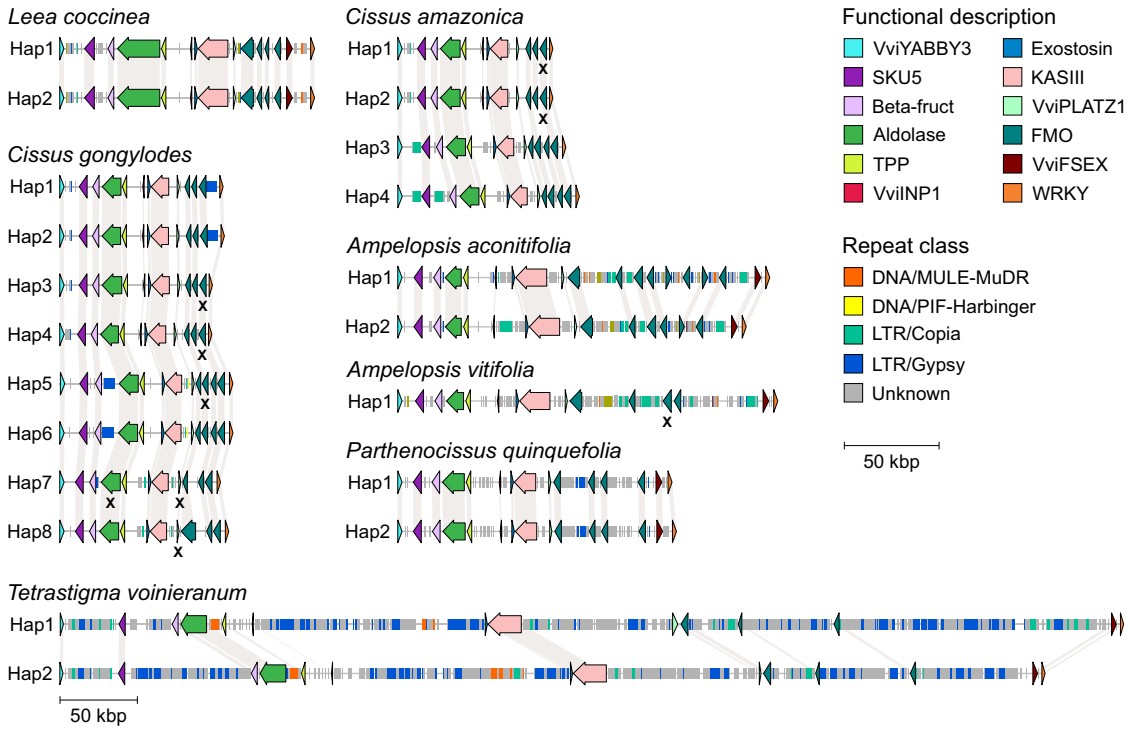

**Fig. 3 | Schematic representations of the gene and intergenic repeat contents in the homologous regions of the *Vitis* sex-determining region in Vitales.** Each haplotype is represented with arrows to depict annotated genes. Genes affected by nonsense mutations are indicated with an X. The scale in legend denotes the length of the region for all the haplotypes except *T. voinieranum* ones. Hap haplotype. Source data are provided as a Source Data file.

*voinieranum* (687.5 and 637.0 kbp). In the outgroup *L. coccinea*, both haplotypes were identical with a size of 134.6 kbp.

The variation in the VSR sizes correlated with differences in intergenic repeat content across Vitaceae clades (Fig. 3 and Supplementary Data 7), suggesting that this is the primary factor driving length differences between haplotypes. In the three *Ampelopsis* haplotypes, repetitive elements accounted for $64.4 \pm 9.3$ kbp, with $13.6 \pm 3.7$ kbp attributed to LTR *Copia* elements. By contrast, the *Cissus* species exhibited low repeat content, averaging $7.7 \pm 4.3$ kbp, with notable differences between haplotypes. For instance, haplotypes 1 and 2 of *C. amazonica* contained 4.0 and 8.4 kbp of LTR *Copia*, respectively, while 5.6 kbp of LTR *Gypsy* elements were identified in haplotypes 1, 2, 5, and 6 of *C. gongylodes*. In *P. quinquefolia*, intergenic repeats comprised 31.7 kbp and 33.6 kbp in haplotypes 1 and 2, respectively. The greatest amount of intergenic repeats was observed in *T. voinieranum*, with 567.1 and 506.2 kbp in haplotype 1 and 2, respectively. In the outgroup *L. coccinea*, intergenic repeats accounted for 21.6 kbp.

## The *Vitis* SDR-homologous region shows no evidence of linkage constraint in dioecious *Tetrastigma* species

Given that *Tetrastigma* spp. are dioecious with an unknown sex determination system, and two VSRs were found in the genome of *T. voinieranum*, we investigated whether the VSR could also be associated with sex determination in *Tetrastigma*. To test this, DNA-seq short reads from thirteen *Tetrastigma* species were retrieved from NCBI and aligned onto the haplotype 1 of *T. voinieranum* (Supplementary Data 6). A total of 3,326,648 SNP positions were identified, of which 15.75% (524,113) were in protein-coding exons. Phylogenetic analysis based on this SNP dataset grouped these species into three clades (Supplementary Fig. 4), consistent with previous study[27]. The SNP dataset was then used to evaluate the LD in *Tetrastigma* spp. LD declined rapidly, with half of the maximum average $r^2$ at 118 bp (Supplementary Fig. 5). Within the VSR, 2635 SNP positions were identified

across the thirteen *Tetrastigma* species, with each species contributing an average of $326.6 \pm 176.2$ SNPs. The majority of SNPs (61.2%) were located within genes, and 27.2% were shared by at least two species (Supplementary Fig. 6). However, only 63,490 bases, representing 9.23% of the VSR, were covered by at least one sequencing read in all thirteen species. This limited coverage likely reflects a combination of low genome coverage (< 3.9×) and substantial sequence divergence across species, both of which may have constrained variant detection. The ratio of homozygous to heterozygous SNPs was significantly higher in *Tetrastigma* species compared to *M. rotundifolia* and *Vitis* spp. (Supplementary Fig. 7; Kruskal–Wallis test followed by post hoc Dunn's test; adjusted *P* value < 0.05), indicating lower heterozygosity in *Tetrastigma* within the VSR. In established SDRs, elevated heterozygosity is typically observed due to the accumulation of sex-specific alleles, as recombination suppression limits genetic exchange between homologous chromosomes. The comparatively low heterozygosity observed in the *Tetrastigma* VSR suggests an absence of sex-specific divergence in this region, consistent with the lack of a differentiated SDR. To further investigate, we analyzed LD patterns in the VSR to evaluate potential recombination suppression, a hallmark of SDRs in dioecious species. Although the flower sex of *T. voinieranum* and the thirteen *Tetrastigma* species is unknown, strong LD would be expected in the SDR if recombination suppression were present, given that *Tetrastigma* is a dioecious genus. However, no such pattern was observed: the average $r^2$ was $0.005 \pm 0.035$ per kbp across the VSR (Fig. 4a and Supplementary Fig. 8), and $0.030 \pm 0.081$ per kbp when considering only 1-kbp windows with at least five SNPs. These findings suggest that the VSR in *T. voinieranum* is likely not subject to recombination suppression. Nonetheless, this conclusion should be interpreted with caution due to the limited sequence coverage and the availability of SNP data only across species.

In addition to evaluating LD, we analyzed sequence divergence between the two VSR haplotypes of *T. voinieranum*. In SDRs, the absence of recombination often leads to increased synonymous

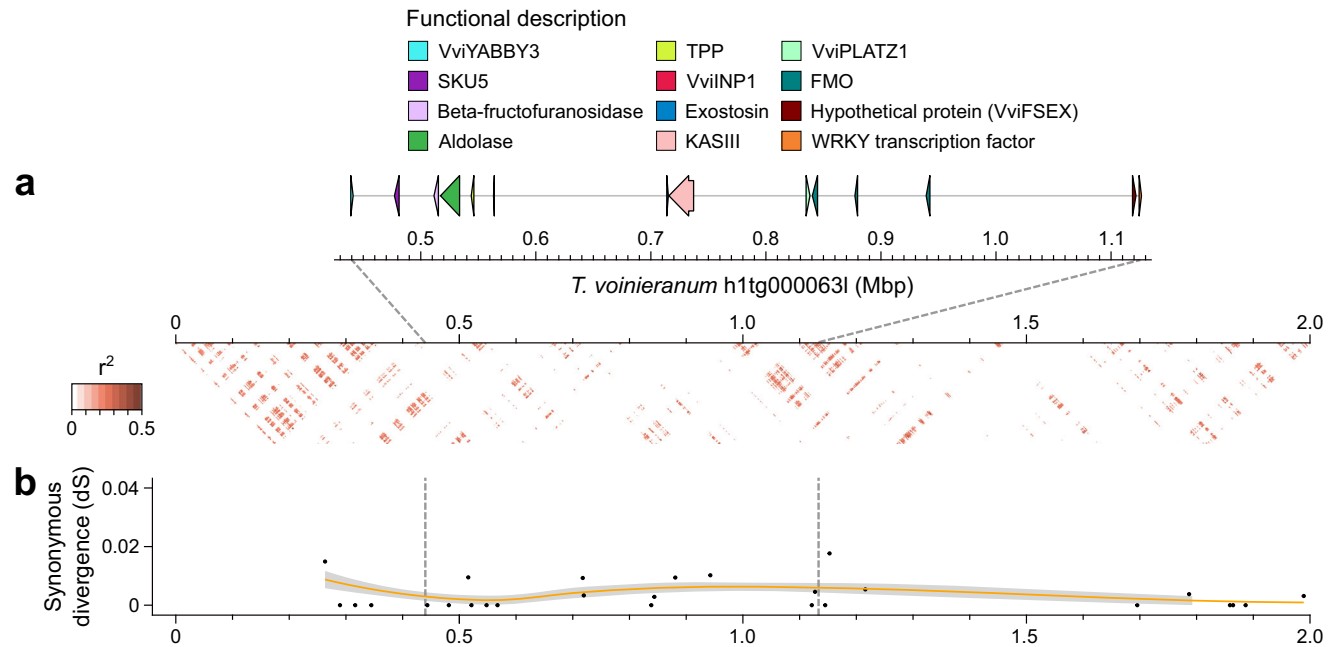

**Fig. 4 | Homologous region of the *Vitis* sex-determining region in *T. voinieranum*. a** Linkage disequilibrium as the mean $r^2$ per kbp. **b** Synonymous divergence (dS) between the two haplotypes containing the *Vitis*-homologous sex-determining region of *T. voinieranum*. The orange line represents the loess-smoothed segmented linear model of the dS with a 95% confidence interval. Source data are provided as a Source Data file.

divergence (dS) between M and F haplotypes, as genetic material is exchanged less frequently[28]. The synonymous divergence (dS) between the two VSR haplotypes of *T. voinieranum* was low and did not show a significant increase in the VSR region (Fig. 4b; Kruskal–Wallis test; *P* value = 0.56). This stable dS in the VSR further supports the absence of recombination suppression in this region. Taken together, these results suggest that the VSR in *T. voinieranum* is probably not associated with sex determination.

**The sex-determining region of *Muscadinia rotundifolia***
In a previous study, we determined the gene content of the two VSR haplotypes of the male *M. rotundifolia* Trayshed[4]. However, the boundaries of the SDR in *M. rotundifolia*, and its candidate sex-determining genes remain unknown. To define the boundaries of the muscadine SDR, we first aligned two genetic sex-linked markers[21] on the diploid chromosome-scale genome of *M. rotundifolia* Trayshed[29]. These markers were found at 3,961,704 and 4,493,503 bp on the F haplotype of Trayshed located on chromosome 2 haplotype 2. Next, we assessed LD in *M. rotundifolia* using whole-genome sequencing data from ten muscadine individuals, three females, three males, and four hermaphrodites (Supplementary Data 6). LD decayed rapidly, with half of the maximum average $r^2$ observed at 2.3 kbp (Supplementary Fig. 9), consistent with findings from previous study in grapes[30]. Patterns of LD on haplotype 2 of Trayshed's chromosome 2 revealed elevated LD between 4.268 and 4.414 Mbp, with an average $r^2$ of $0.60 \pm 0.18$ per kbp. Consequently, the muscadine SDR in Trayshed spans ~146 kbp for the F haplotype, from the vacuolar processing enzyme (VPE) gene to the third *FMO* gene (16 protein-coding genes), while its M haplotype is ~232 kbp and contains 15 protein-coding genes (Fig. 5a and Supplementary Fig. 10). In *Vitis* spp., a high LD was observed between *VviYABBY3* and *VviAPT3*[5]. These results suggest that the boundaries of the *Vitis* and *M. rotundifolia* SDRs are highly comparable.

To investigate the structure of the muscadine SDR, we sequenced and assembled the genomes of the accessions Fry (female) and DVIT1750 (male) using PacBio continuous long reads DNA sequencing (Supplementary Data 3 and 4). The diploid genome assemblies of Fry

and DVIT1750 were 700.9 Mbp and 761.4 Mbp in length, respectively (Supplementary Data 5), consistent with the genome size of Trayshed (825 Mbp)[4]. Thanks to the high contiguity of these diploid genomes, we identified the four SDR haplotypes in Fry and DVIT1750. The assignment of these haplotypes to either the F or M haplotype was based on the structure of Trayshed's haplotypes[4]. Synonymous divergence between the M and F haplotypes of *M. rotundifolia* Trayshed and DVIT1750 was significantly greater at the SDR compared to the adjacent regions (Kruskal–Wallis test; *P* value < 0.05), while no significant difference was observed between the two F haplotypes of Fry (Supplementary Fig. 11). These findings support that the muscadine SDR is under recombination suppression.

Alignment of all muscadine haplotypes to Trayshed's F haplotype revealed that the four F haplotypes were highly similar in structure, as were the two M haplotypes from Trayshed and DVIT1750 (Fig. 5b). Both M haplotypes exhibited an inversion of approximately 209 kbp relative to the F haplotype, spanning from 4,506,939 to 4,715,941 bp in the M haplotype of Trayshed. Based on the genetic distances between the F and M haplotypes at both ends of the inversion, we estimated that the structural variation occurred $40.3 \pm 13.2$ Mya. The breakpoints of this inversion closely aligned to the boundaries of strong LD (Fig. 5a). The region spanned 255.3 and 242.9 kbp in the M haplotypes of Trayshed and DVIT1750, respectively, whereas the F haplotypes averaged $146.8 \pm 1.3$ kbp. This size difference was largely attributed to the higher content of repetitive elements in the intergenic space of the M haplotypes compared to their F counterparts (Fig. 5c).

Repetitive elements accounted for $23.6 \pm 0.9$ kbp in the F haplotypes and $103.7 \pm 2.0$ kbp in the M haplotypes (Supplementary Table 1). For example, multiple LTRs were observed in the intergenic region of the M haplotypes, specifically upstream the *VPE* gene, between *VviPLATZ1* and the gene encoding the 3-ketoacyl-acyl carrier protein synthase III (KASIII). We estimated the insertion of LTRs between *VviPLATZ1* and *KASIII* in the M haplotypes to have occurred $41.7 \pm 10.3$ Mya based on their genetic distance. Additionally, several *Mutator*-like elements (MULEs) were found in the intergenic region between *VviYABBY3* and the NAC TF gene in the M haplotypes, which were absent in the F haplotypes. Despite these differences, the gene content

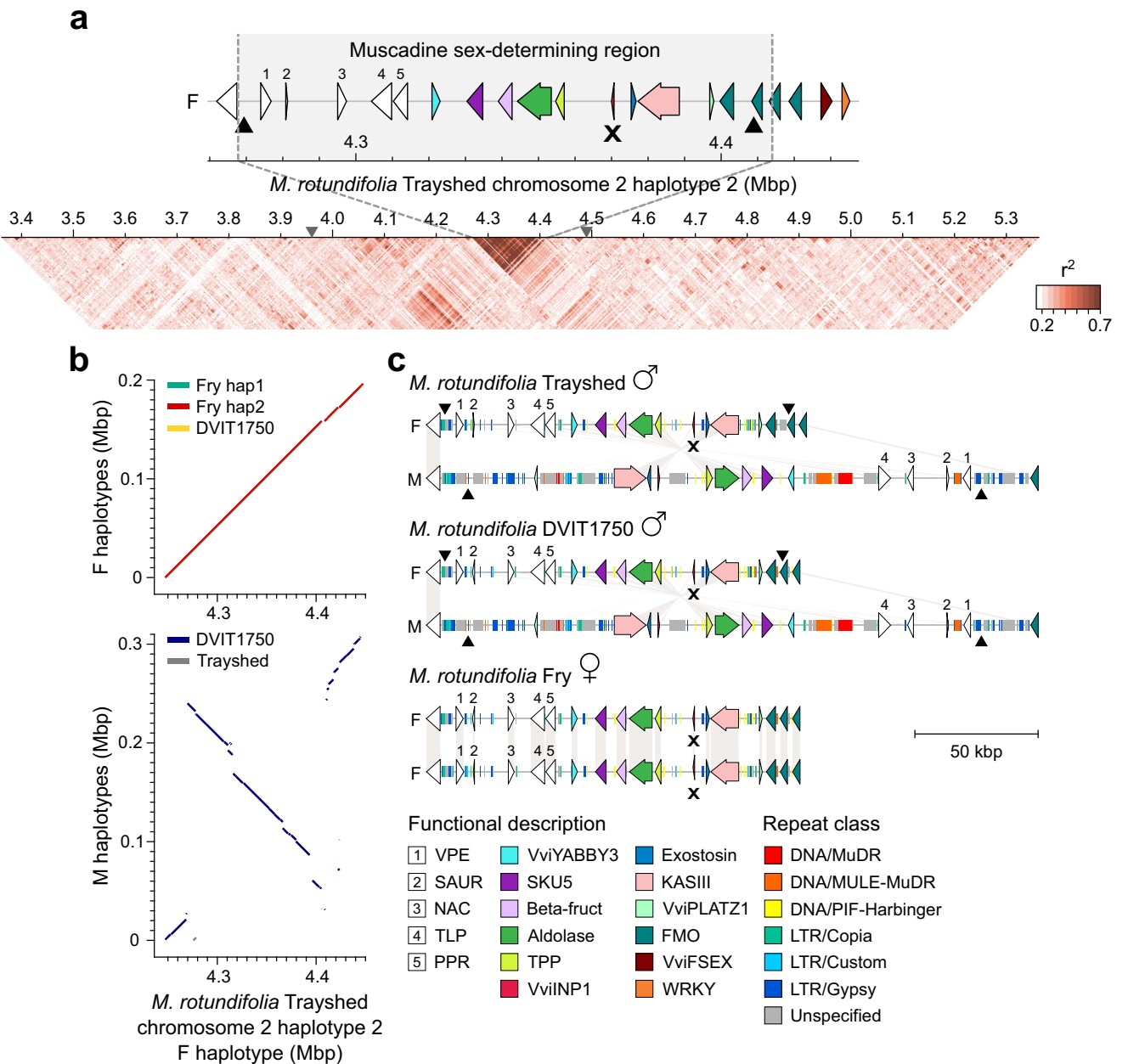

**Fig. 5 | Sex-determining region of *M. rotundifolia*. a** Linkage disequilibrium as the mean $r^2$ per kbp. The gray triangles on the scale mark the position of the sex-linked markers in muscadines, S2_4635912 and S2_5085983[21]. **b** Whole-sequence alignments of the F (top) and M (bottom) haplotypes against the F haplotype of *M. rotundifolia* Trayshed. **c** Schematic representations of the gene and intergenic repeat contents in the sex-determining region (SDR) of the males *M. rotundifolia* Trayshed and DVIT1750, and the female muscadine Fry. The symbols ♀ and ♂ represent the female and male flower sex type of the muscadine individuals. For each haplotype, genes are represented with arrows and repeats with rectangles. White arrows depict genes that are not part of the *Vitis* SDR. The black triangles mark the position of the breaking points of the inversion in the M haplotype compared to the F haplotype. Pseudogenes are indicated with an X. Gene content representations from (**a**) and (**c**) share the same legend. Source data are provided as a Source Data file.

in the region was highly conserved between the F and M haplotypes (Fig. 5b). The 8-bp deletion in *VviINP1* was present in all F haplotypes and was homozygous in the female Fry, similar to the female *Vitis* individuals.

To estimate when recombination ceased in the grape SDR of *Vitis* and *M. rotundifolia*, we calculated the synonymous divergence (dS) between the M and F allele of each gene within the region common to the *Vitis* and muscadine SDRs, i.e., from *VviYABBY3* to *VviPLATZ1*, for each male individual. The dS values varied considerably along the SDR in both *Vitis* and muscadines (Supplementary Fig. 12). In *Vitis* spp., the dS ranged from 0.0034 to 0.0543, while in *M. rotundifolia*, the range was higher, from 0.0197 to 0.064.

These results suggest that recombination totally ceased slightly earlier in *M. rotundifolia* (20.3 ± 8.4 Mya) compared to *Vitis* spp. (13.8 ± 8.5 Mya).

Comparison of the proteins encoded by the genes within the muscadine SDR revealed a sex-linkage pattern from the *VPE* gene to the *FMO* gene (Supplementary Fig. 13). When compared with the SDR proteins from *Vitis* spp., nine out of fourteen proteins showed a genus-specific separation (Supplementary Fig. 14). We observed clustering of the F alleles for *VviINP1* and exostosin genes in both *Vitis* and *M. rotundifolia* relative to the M and hermaphrodite-associated (H) alleles, though no clustering of the M alleles compared to the F and H alleles was observed for any gene.

The two M alleles of *M. rotundifolia* were distinctly separated from all the other alleles for three proteins: VPE, the thylakoid luminal protein, and the aldolase. Next, we compared the promoter region (up to 3 kbp upstream of the transcription start site (TSS)) of each SDR gene (Supplementary Fig. 15). Similar to the protein level analysis, a sex-linkage pattern was detected for the F alleles of *VviINP1*, exostosin, and *VviPLATZ1* relative to their M and H counterparts. *VviPLATZ1* has been shown to influence grapevine stamen erectness[6], suggesting that *VviPLATZ1* may play a similar role in *M. rotundifolia*. Notably, the only clear separation between M alleles and the F/H alleles was observed in the promoter region of *VviYABBY3* (Fig. 6a and Supplementary Fig. 15). Sequence alignment of the *VviYABBY3* promoter region revealed fifteen sites specific to the M alleles of *Vitis* and *M. rotundifolia* (Supplementary Fig. 16). To assess the potential impact of the promoter sequence variation of *VviYABBY3* on gene expression regulation, we compared the TF-binding sites. A MYB59-binding site, located $2104 \pm 14$ bp upstream of the TSS (from 2366 to 2373 bp on the alignment (Supplementary Fig. 16), was found specific to both *Vitis* and *M. rotundifolia* M alleles, i.e., present in all M haplotypes but absent in all F and H haplotypes. Additionally, a greater number of TF-binding sites in M alleles compared to F and H alleles was identified for seven other TFs (Supplementary Data 8), including MYB81, which played a critical role in the developmental progression of microspores in *Arabidopsis thaliana*[31], and HOMEODOMAIN GLABROUS 1, involved in floral identity ref. [32]. In contrast, F and H alleles contained more TF-binding sites for five TFs, such as SPL4 and SPL12, two squamosa promoter-binding protein-like proteins associated with flowering in *A. thaliana*[33,34].

To investigate whether the two alleles of *VviYABBY3* are expressed in a sex-specific manner, we quantified their gene expression in the sexual structures (ovaries and stamens) of the male accessions DVIT1750 and Trayshed, and the female Fry (Fig. 6b). In the ovaries, the F allele of *VviYABBY3* was highly expressed ($107.2 \pm 23.5$ transcripts per million (TPM)) in the female flowers of Fry, while its expression was low in the male flowers of DVIT1750 and Trayshed ($2.6 \pm 0.6$ TPM). However, the M allele of *VviYABBY3* was more highly expressed in the ovaries of both male muscadines compared to the F allele (Kruskal–Wallis test; $P$ value $< 0.05$), suggesting a potential role in suppression of femaleness. Regarding the stamens, both alleles were lowly expressed in the three accessions (Fry, $0.2 \pm 0.1$ TPM; DVIT1750, $1.0 \pm 0.8$; Trayshed, $1.6 \pm 0.2$), with the M allele of *VviYABBY3* more highly expressed than the F allele in DVIT1750 (Kruskal–Wallis test; $P$ value $< 0.05$), suggesting a potential role in maleness.

In summary, defining the boundaries of the *M. rotundifolia* SDR revealed that the muscadine and *Vitis* SDRs share similar boundaries. The generation of additional *M. rotundifolia* genomes confirmed the presence of an inversion in the M haplotype relative to the F haplotype. It also demonstrated that the 8-bp deletion in *VviINP1* is homozygous in female muscadines, suggesting that *VviINP1* could also function as the male-determining gene in muscadines. Sequence analysis of the SDR genes confirmed the absence of an M-specific mutation shared between *Vitis* and *M. rotundifolia*, whereas M-specific mutations affecting TF-binding sites were identified in the promoter region of *VviYABBY3*. As in *Vitis* spp., the M allele of *VviYABBY3* was more highly expressed in the ovaries of male flowers compared to the F allele, and the F allele in male flowers exhibited lower expression than in female flowers. This suggests that the M allele of *VviYABBY3* may also function as the female-suppressing factor in muscadines. Altogether, our results indicate that the *Vitis* and *M. rotundifolia* SDRs not only share similar boundaries but also harbor similar candidate sex-determining genes, suggesting a common evolutionary origin.

## Discussion

The evolution of sex determination in plants, particularly in dioecious species, is a complex process shaped by genetic changes in regions comprising genes associated with female and male organogenesis[35]. In garden asparagus (*Asparagus officinalis*), the gene *SUPPRESSOR OF FEMALE FUNCTION* (*SOFF*) is associated with the malformation of the stylar tube and receptive stigma, while *DEFECTIVE IN TAPETAL DEVELOPMENT AND FUNCTION1* (asp*TDF1*) is involved in anther development and pollen production[36]. In kiwifruit (*Actinidia deliciosa*), the female-suppressing gene, *Shy Girl* (*SyGI*), represses the development of the pistil through negative regulation of the cytokinin signaling, and the fasciclin-like gene *Friendly Boy* (*FrBy*) is involved in proper tapetum degradation[37]. Concerning the *Vitis* SDR, at least five genes within the locus, including the two candidate sex-determining genes *VviYABBY3* and *VviINP1*, are, or have homologous genes, involved in floral organ development and identity in plants. Mutations in homologous genes of the *Vitis* SDR locus in Arabidopsis have been shown to cause significant alterations in flower morphology and fertility (Table 1), highlighting the functional importance of these genes in proper floral development.

The high conservation and collinearity of the genes within the *Vitis* SDR locus among angiosperms suggest that this region has been under strong evolutionary pressure to maintain its structure and function (Fig. 2)[38]. Investigating whether genes flanking the SDR also exhibit collinearity, and how far this conservation extends, could provide insight into whether additional gene functions beyond sex determination are conserved across flowering plants. The higher rank-normalized gene conservation scores further imply that not only the arrangement but also the sequence of these genes is highly preserved, reinforcing the idea that this locus likely plays a critical role in flowering processes[39]. The absence of a region collinear to the *Vitis* SDR in non-flowering plants further supports its potential specialized role in angiosperm reproductive development[40]. So far, the *Vitis* SDR is the only known plant SDR that exhibits high gene collinearity and conservation across angiosperms. In most plant lineages, SDRs evolve rapidly following recombination suppression, often resulting in large-scale degeneration of the M haplotype and hemizygosity for many genes[41]. Consequently, gene content in the M haplotype is rarely conserved even among closely related species. To date, only the male-determining gene *FrBy* in kiwifruit has been shown to have orthologs in 32 angiosperm species, three of which retain similar function, suggesting limited functional conservation across lineages[37]. In contrast, the *Vitis* SDR, despite its estimated age (~ 20 Mya), shows minimal gene content divergence between F and M haplotypes. This unusual degree of conservation suggests that the *Vitis* SDR may contain genes essential for core developmental processes such as flowering. Functional studies, including targeted gene knockouts, will be critical to determine whether this constraint underlies the exceptional stability of the *Vitis* SDR. More broadly, investigating whether other plant SDRs originate from similarly conserved genomic regions or whether the *Vitis* SDR represents a unique evolutionary case remains an open and compelling question in the study of plant sex chromosome evolution.

Among Vitaceae, we observed variation in haplotype size mainly due to differences in the repeat content of intergenic regions, with types of repetitive elements varying between species. Although most of the genes within the locus were conserved, some differences in gene content were evident at the clade, species, and individual levels. In the outgroup *L. coccinea*, two VSR haplotypes were identical, indicating that the region is homozygous. Across Vitaceae, the size of the VSRs varied significantly between clades, ranging from 80.2 kbp in *C. amazonica* to 687.5 kbp in *T. voinieranum* (Fig. 3).

Because the sex determination system of the dioecious genus *Tetrastigma* is unknown, and *T. voinieranum* genome contained two VSRs, we investigated if the region could also be associated with sex determination. A requisite for the SDR formation is the linkage constraint between the two sex-determining genes[42]. Evaluation of the LD in the region using short DNA-sequencing reads from thirteen *Tetrastigma* species showed a low LD, and no pattern of recombination suppression could be observed. Low LD indicates that the region is

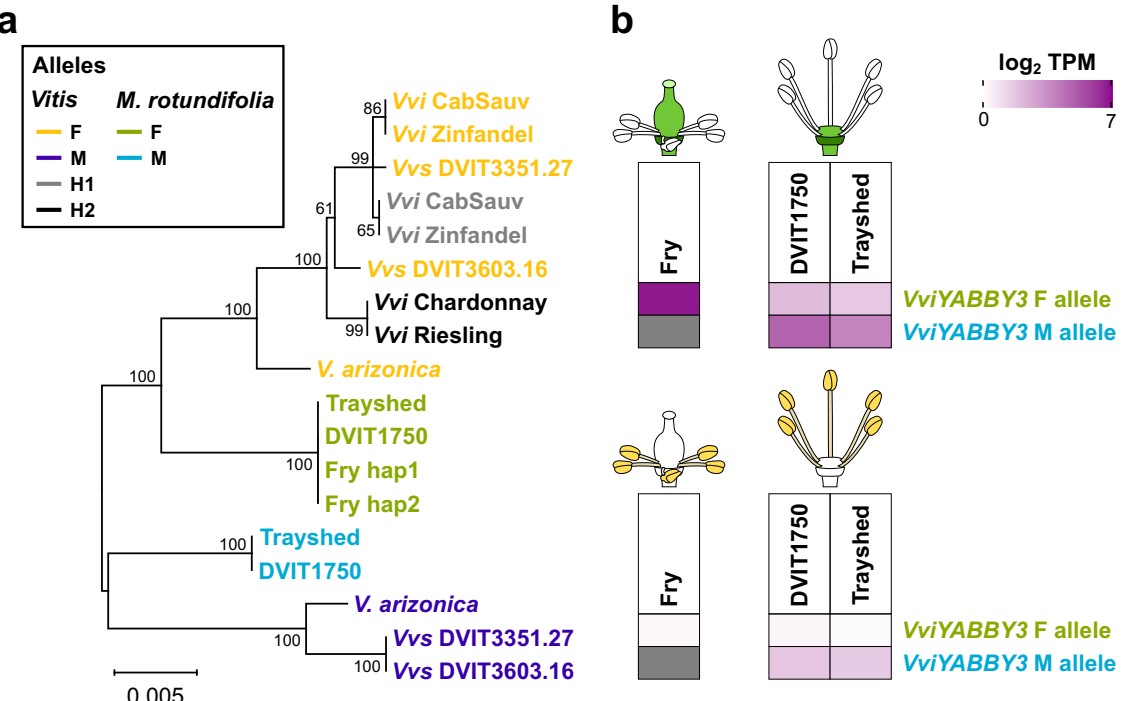

**Fig. 6 | Differences between the F and M allele of *VviYABBY3* in *M. rotundifolia*. a** Phylogenetic tree of the promoter region of *VviYABBY3* in *Vitis* and *M. rotundifolia*. **b** Gene expression of the F and M alleles of *VviYABBY3* in flower ovaries and stamens represented as the log2-transformed mean of the transcripts per million (TPM) of the three biological replicates (*n* = 3). Source data are provided as a Source Data file.

unlikely associated with sex determination, suggesting that dioecy emerged in two distinct regions in the Cayratieae and Viteae clades. However, the genome coverage of the *Tetrastigma* species retrieved from NCBI was low (< 3.9×; Supplementary Data 6). Although we limited variant calling to species with at least 1.9× coverage, only 63,490 bases out of 687.5 kbp were covered by at least one sequencing read in all thirteen species. This limited coverage, along with interspecific sequence divergence, likely constrained both variant calling and LD analysis. To address these limitations, higher-quality, species-specific sequencing data from additional *Tetrastigma* species and populations will be necessary to confirm the absence of linkage constraints in this region. When combined with flower sex phenotyping, such data could facilitate the identification of the SDR in *Tetrastigma*.

From evolutionary studies, the divergence between *Vitis* and *Muscadinia* genera was estimated between 18 and 47 Mya[17–19]. In this study, we aimed to better characterize the muscadine SDR to assess the commonalities and differences between the two grape SDRs. The identification of the muscadine SDR using sex-associated genetic markers and pattern of LD showed that the SDR of *M. rotundifolia* was located in a region similar to the *Vitis* SDR. This suggests that dioecy emerged before the divergence of the two grape genera. Additional sequencing data from *M. rotundifolia*, such as recombinants, will be necessary to confirm and narrow down the muscadine SDR boundaries. This same approach was successful in determining the boundaries of the *Vitis* SDR[5]. As a result of breeding efforts, two independent hermaphroditic lines, designated H1 and H2, have been identified in muscadines[43]. Comparing the SDR from additional male, female, and hermaphrodite *M. rotundifolia* would thus help to refine the boundaries of the region and better understand how hermaphroditism arose. A large inversion was found in the M haplotype of both male muscadines relative to the F counterpart. In *Carica papaya*, two large inversions were also observed in the male- and hermaphrodite-specific region of the Y chromosome compared to the X region[44,45]. Both inversions were found to have caused recombination suppression in

the papaya SDR, suggesting that the inversion in the M haplotype of *M. rotundifolia* could play a similar role in the muscadine SDR. However, we estimated that the inversion occurred 40.3 ± 13.2 Mya, which overlaps with the divergence time between *Vitis* and *Muscadinia*. Another hypothesis is that the inversion in the M haplotype of *M. rotundifolia* occurred after the divergence between the two grape genera to further suppress recombination in the locus. This would explain why recombination completely stopped slightly earlier in *M. rotundifolia* (20.3 ± 8.4 Mya) compared to *Vitis* spp. (13.8 ± 8.5 Mya).

Regarding the candidate sex-determining genes of *M. rotundifolia*, some evidence suggest that they could be the same genes that the ones previously discovered in *Vitis* spp.[4,5]. Like in *Vitis* spp., a homozygous 8-bp deletion was found in *VviINP1* in the female *M. rotundifolia* Fry. Pollen grains of female flowers from muscadines, including the accession Fry, were reported as acolporated and sterile[20]. This suggests that mutations in *VviINP1* could also be the female-determining factor in muscadines. Regarding the female-suppressing factor, the promoter of *VviYABBY3* was the only sequence showing a M linkage in both *M. rotundifolia* and *Vitis* genera (Fig. 6a). Sequence differences between the M and F alleles of *VviYABBY3* promoter region were found to affect the potential TF-binding sites in terms of TF and site numbers, suggesting a potential impact on the transcriptional regulation of the two alleles. In addition, the M allele of *VviYABBY3* was detected as more highly expressed compared to its F counterpart in ovaries (Fig. 6b), raising the possibility that this allele may act as the female-suppressing gene in *M. rotundifolia*. However, no M-specific amino acid differences were found in the M allele of VviYABBY3 in *Vitis* and *M. rotundifolia*. Further investigation will be necessary to determine whether female suppression in the two grape genera is mediated by shared molecular mechanisms, and to clarify the transcriptional regulatory basis underlying the differential expression of the M and F alleles of *VviYABBY3*. Because the SDRs of *Vitis* spp. and *M. rotundifolia* are located in a similar region, contain similar genes, and share candidate sex-determining genes, this suggests that a common evolutionary

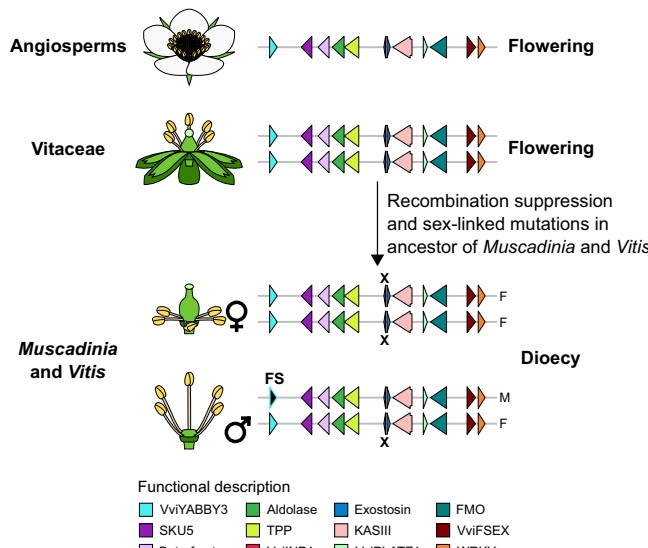

**Fig. 7 | Model of the establishment of the *Vitis* and *Muscadinia* sex-determining regions in a highly conserved and flowering-associated region in angiosperms.** Schematic representations of the gene content in the homologous regions of the *Vitis* sex-determining region in angiosperms, Vitaceae, and *Muscadinia* and *Vitis* genera. The symbols ♀ and ♂ represent the female and male flower sex types. Arrows depict genes. The nonsense mutation in the candidate male-determining in grapes, *VviINP1*, is indicated with an X. The candidate female-suppressing (FS) gene *VviYABBY3* is also indicated.

origin of the SDRs in these two grape genera. Finally, alignment of DNA-seq short reads from 159 Vitaceae accessions, showed that the mutation was absent in all accessions, including the two other *Viteae* genera, *Ampelocissus* and *Pterisanthes* (Supplementary Data 7). This suggests that the candidate male-sterility mutation is specific to the *Vitis* and *Muscadinia* genera. However, DNA sequencing of additional samples from the hermaphrodite Vitaceae genera *Ampelocissus*, *Pterisanthes*, and *Nothocissus* will be necessary to confirm it.

Finally, our results support a model where, in an ancestor of *Vitis* and *Muscadinia*, mutations within a highly conserved region associated with flowering led to the emergence of dioecy in this lineage (Fig. 7). Our results suggest that dioecy emerged through the evolution of a genomic region which gene content is highly collinear and conserved in angiosperms and composed of multiple genes playing a role in flower development and morphology, and sexual fertility.

## Methods
### Plant material
For genome sequencing, young leaves were collected from eighteen accessions: two *M. rotundifolia* individuals, Fry and DVIT1750, six *Ampelopsis* individuals representing four species, six *Cissus* species, one *P. quinquefolia* DVIT2400, *T. voinieranum*, one *Causonis japonica*, and *L. coccinea* 1464. Information about the sequenced accessions is provided in Supplementary Data 3. For RNA-sequencing, inflorescences from *M. rotundifolia* Fry (female), Trayshed (male), and DVIT1750 (male), were collected at full flowering with 50% caps off (E-L 23)[46]. Ovaries and stamens from cap-off flowers were then sampled to represent three biological replicates per accession. All plant material was immediately frozen after collection and ground to powder in liquid nitrogen.

### DNA and RNA extraction, library preparation, and sequencing
For long-read DNA sequencing, high molecular weight genomic DNA (gDNA) was isolated as in ref. 47 from *M. rotundifolia* Fry and DVIT1750, *A. aconitifolia* DVIT2492, *A. glandulosa* var. *brevipedunculata* PI

597579, *C. amazonica*, *C. gongylodes*, *P. quinquefolia* DVIT2400, *T. voinieranum*, and *L. coccinea* 1464. DNA purity and quantity were evaluated using a Nanodrop 2000 spectrophotometer (Thermo Scientific, IL, USA) and a Qubit™ 1× dsDNA HS Assay Kit (Thermo Fisher, MA, USA). DNA integrity was assessed with the DNA High Sensitivity kit (Life Technologies, CA, USA) and by pulsed-field gel electrophoresis. SMRTbell libraries of *M. rotundifolia* Fry and DVIT1750, *A. aconitifolia* DVIT2492, *A. glandulosa* var. *brevipedunculata* PI 597579, *C. amazonica*, *C. gongylodes* were prepared with 15 μg of sheared DNA using the SMRTbell Express Template Prep Kit (Pacific Biosciences, CA, USA) following the manufacturer's instructions, size-selected using a Blue Pippin instrument (Sage Science, MA, USA) with a cutoff size range of 20–80 kbp, and cleaned with 1× AMPure PB beads. For the HiFi libraries of *P. quinquefolia* DVIT2400, *T. voinieranum*, and *L. coccinea* 1464, 12 μg of high molecular weight gDNA from each accession was sheared to a size of 15–20 kbp using the Megaruptor (Diagenode, Denville, NJ, USA). HiFi libraries were prepared using the SMRTbell Express Template Prep Kit 3.0 (Pacific Biosciences, CA, USA), size-selected with a cutoff size range of 10–50 kbp using using a BluePippin (Sage Sciences, MA, USA), and cleaned using 1× AMPure PB beads. Concentration and size distribution of the libraries were assessed using a Qubit™ 1× dsDNA HS Assay Kit (Thermo Fisher, MA, USA) and Femto Pulse System (Agilent, CA, USA), respectively. SMRTbell and HiFi libraries were sequenced using PacBio Sequel II system (Pacific Biosciences, CA, USA) at the DNA Technology Core Facility, University of California, Davis (Davis, CA, USA). Summary statistics of long-read DNA sequencing are provided in Supplementary Data 4. For short-read DNA sequencing, DNA extraction and library preparation were performed as ref. 4. Regarding the RNA-sequencing from the muscadine flowers, RNA extraction and library preparation were performed as in ref. 48. RNA extraction for one bioreplicate of stamens from Trayshed flowers failed and tissue from the bioreplicate was not sufficient to repeat the extraction. DNA and cDNA libraries were sequenced using an Illumina HiSeqX Ten system (IDseq, Davis, CA, USA) as 150-bp paired-end reads and an Element Bio AVITI as 80-bp paired-end reads, respectively.

### Genome assembly and annotation
Genome assembly from PacBio CLR reads was performed with FALCON-Unzip[47] as ref. 4 and with hifiasm[49] as ref. 50 for the HiFi reads. For the muscadine genomes, repetitive elements were identified using RepeatMasker v.open-4.0.6 (http://www.repeatmasker.org) and a grape repeat library[51]. For the genome assemblies of the *Ampelopsis*, *Cissus*, *Parthenocissus*, *Tetrastigma*, and *Leea* species, repeat libraries were created separately with RepeatMasker v.open-4.0.6[52] and RepeatModeler2 v.2.04[53] starting from the RepBase library v.20160829 and Cabernet Sauvignon custom repeat library[54]. For the gene annotation, EVidenceModeler v.1.1.1[55] was used to integrate the ab initio predictions of SNAP v.2006-07-28[56], Augustus v.3.0.3[57], GeneMark-ES v.4.321[58], the gene models identified by PASA v.2.1.0[59] with experimental evidence, and the gene models detected by Exonerate v.2.2.0[60] using the proteins from Swissprot viridiplantae (downloaded on 2016.03.15). As experimental evidence, we used the protein-coding sequences and the proteins of the isoforms identified using IsoSeq data from Cabernet Sauvignon[54], North American *Vitis* species[30], and *M. rotundifolia* Trayshed[61], as well as the transcriptome of *V. vinifera* ssp. *vinifera* Corvina[54]. Sequence redundancy was reduced using CD-HIT v.4.6[62] with the parameters "cd-hit-est -c 0.95". Gene space completeness was evaluated using BUSCO v.5.4.7 and the library embryophyte_odb10[63].

### Phylogenetic analysis
Phylogenetic tree of the thirteen Vitaceae and *L. coccinea* (Fig. 1) was based on the 353 putatively single-copy protein-coding genes in angiosperms[64]. Orthologues of the proteins encoded by the *Arabidopsis thaliana* genes from the Angiosperms353 set[64] were identified

among the fourteen genomes. Proteins of *V. vinifera* ssp. sylvestris O34-16 (female) and DVIT3351.27 (male), were retrieved from ref. 4, *V. berlandieri* 9031 and *V. girdiana* SC2 from ref. 30, and *M. rotundifolia* Trayshed from ref. 29, respectively. Proteins of *A. thaliana* were aligned and annotated on the primary and haplotype 1 contigs using miniprot v.0.4-r174-dirty[65]. For each protein, the alignment with the greatest coverage and identity, with no frameshift nor stop codon, was retained. Protein sequences were generated using gffread from Cufflinks v.2.2.1[66]. Multi-sequence alignments of the proteins were performed with MUSCLE v.3.8.31[67]. Alignments were then concatenated and parsed using Gblocks v.91b[68] with up to 8 contiguous non-conserved positions, a minimum length of a block of 10, and no gap. Phylogenetic analysis was conducted with MEGAX[69] using the Maximum Likelihood method, the JTT matrix-based substitution model[70] with four Gamma categories, and 1000 replicates. For inferring the divergence times between genera, the RelTime method[71] was applied using three calibration points corresponding to the divergence time between (i) the *Vitis* and *Muscadinia* spp. (41.58 million years ago), (ii) the Viteae and Parthenocisseae clades (59 million years ago), (iii) the *Tetrastigma* and *Cissus* spp. (72.75 million years ago)[72], with a normal distribution and a standard deviation of 1.

Other phylogenetic analyses were performed with MEGAX[69] using the Neighbor-Joining method[73] and 1000 replicates. The Poisson correction method[74] and the Kimura 2-parameter method[75] were used for generating the trees based on protein and DNA sequence comparison, respectively. Prior the phylogenetic tree of the thirteen *Tetrastigma* species, a sequence alignment matrix was created from the SNPs identified between the thirteen species and the haplotype 1 of *T. voinieranum* genome using the script vcf2phylip.py v.2.0[76].

### Gene collinearity analysis

Gene collinearity was evaluated between the haplotype 1 of Cabernet Sauvignon genome[4] and 56 plant genomes. Information about the genome and annotation files used for this analysis can be retrieved in Supplementary Data 1.

To identify orthologous regions, primary proteins of each plant genome haplotype were aligned in a pairwise fashion against each other using DIAMOND blastp v2.0.13.151[77] with the following parameters: "-max-target-seqs 100 -evalue 0.001". Collinear gene blocks were identified with MCScanX v.11.Nov.2013[22] and the parameters "-s5 -m25 -w1". Gene content of the haplotype 1 of Cabernet Sauvignon was split in 2307 windows of 12 consecutive genes. For each plant genome haplotype, the collinear blocks with the highest number of genes from the Cabernet Sauvignon 12-gene windows were selected to form orthologous windows. Orthologous and paralogous regions were detected using GENESPACE v.1.3.1[24], default parameters, and primary proteins. Detected collinear regions were then compared to Cabernet Sauvignon 12-gene windows to form homologous windows. In each homologous window, the collinear gene with the greatest bit score to each Cabernet Sauvignon gene was selected.

For each pair of collinear genes, a gene conservation score was calculated by dividing the alignment score (bit score) between collinear genes by the bit score of the grape reference protein against itself, resulting in values between 0 and 1[39]. Gene conservation scores were then rank-normalized across eudicots, and the other angiosperm clades, separately, to control for the evolutionary distance between the reference and each angiosperm clade (Supplementary Fig. 17).

Genes absent from the syntenic window corresponding to the *Vitis* SDR were searched using miniprot v.0.4-r174-dirty[65] and the SDR protein sequences of the haplotype 1 of Cabernet Sauvignon.

### Sex-determining region localization and haplotype reconstruction

Homologous regions of the *Vitis* SDR were identified by aligning the protein-coding sequences (CDS) from the SDR of Cabernet Sauvignon Hap1 onto the Vitales genome assemblies with GMAP v.2020-06-01[78]. For *M. rotundifolia*, the closest markers to the SDR, S2_4635912 and S2_5085983[21], were aligned onto *M. rotundifolia* Trayshed haplotype 2 using BLAT v.36 × 2[79]. When the alignments of the SDR-associated sequences were found on multiple contigs, NUCmer from MUMmer v.4.0.0[80] and the "-mum" option were used to determine the overlap between contigs. Then, contigs were reconstructed using HaploMake from the tool suite HaploSync v.1.0[81]. Gene models among the grape SDR-like regions were manually refined using the intron-exon structure of the SDR genes from Cabernet Sauvignon. Schematic representations of the gene content were made using the R package gggenes v.0.4.1 (https://wilkox.org/gggenes/).

### Alignment of short DNA sequencing reads

Because of sequencing depth variability, samples from the muscadines and the *Tetrastigma* species were randomly subsampled to 50 and 30 million reads, respectively, with seqtk sample from the package seqtk v.1.2-r101-dirty (https://github.com/lh3/seqtk) and the parameter -s100 before trimming. Short DNA-seq reads were trimmed using Trimmomatic v.0.36[82] and the following settings: "LEADING:3 TRAILING:3 SLIDINGWINDOW:10:20 MINLEN:33". High-quality paired-end reads were aligned onto their corresponding reference genome using bwa v.0.7.17-r1188[83] (Supplementary Data 6). Alignments were visualized using Integrative Genomics Viewer v.2.4.14[84] to evaluate the zygosity status of the 8-bp deletion in *VviINP1*.

### Variant calling

For *Tetrastigma*, samples retrieved from NCBI with at least 1.9× coverage were used for variant calling. Prior variant calling, PCR and optical duplicates were removed with Picard tools v.2.8 (http://broadinstitute.github.io/picard/). The variant calling was performed using HaplotypeCaller from GATK v.4.2.2.0[85] with the parameters "-sample-ploidy 2 -ERC GVCF". VCF files were combined and genotyped using the programs CombineGVCFs and GenotypeGVCFs from GATK v.4.2.2.0[85] with default parameters. SNPs with a quality higher than 30, a depth of coverage higher than five reads, no more than three times the median coverage depth across accessions, a minor allele frequency higher than 0.05, and no missing data among individuals were filtered with vcftools v0.1.15[86]. For *Tetrastigma* genus, SNPs were filtered using a depth of coverage higher than one read and lower than 15 reads due to the low sequencing coverage (3.1 ± 0.6 X). SNPs were further filtered using the "filter" function of bcftools v.1.9[87] and the options "-e QD < 2.0|FS > 60.0|MQ < 40.0 | MQRankSum <−12.5 | ReadPosRankSum < −8.0|SOR > 3.0". SNPs within the muscadine SDR were used to perform a principal components analysis using Plink v.1.90b5.2[88] to infer the flower sex type of the samples SRR6729328, SRR7819188, SRR7819190, SRR11886267 (Supplementary Fig. 18).

### Linkage disequilibrium analysis

To assess linkage disequilibrium (LD) decay, LD was estimated using Plink v.1.90b5.2[88] and the following command: "plink -vcf name.vcf -double-id -allow-extra-chr -r2 gz -maf 0.05 -ld-window 10 -ld-window-kb 300 -ld-window-r2 0 -out name". SNPs from the muscadines were randomly subsampled using the parameter "-thin-count 800000". LD decay was evaluated using the model from Hill and Weir[89]. To explore the LD landscape of the SDR, we used Tomahawk v.beta-0.7.1 (https://github.com/mklarqvist/tomahawk). SNPs comprised in the 2-Mbp region around the SDR in muscadines and *Tetrastigma* spp. were used as input to calculate the LD with the "calc" function. The $r^2$ values were aggregated into bins of 1 kbp using the "aggregate" function, the parameters "-f r2 -r mean -c 5 -x 2000 -y 2000".

### Whole-sequence alignments and structural variation analysis

Pairwise sequence alignments were performed using NUCmer from MUMmer v.4.0.0[80] and the "-mum" option.

## Divergence time estimation

For each male *Vitis* and *M. rotundifolia* individual, the coding sequences of the M and F alleles of each SDR gene were aligned using MUSCLE v.3.8.31[67]. Synonymous divergence (dS) was calculated using the yn00 program in the PAML package v.4.9[90]. The heatmap of the dS values was generated using geom_tile from the R package Tidyverse 2.0.0[91]. To estimate the insertion time of LTR retrotransposons located between *VviPLATZ1* and *KASIII* in the M haplotypes of *M. rotundifolia*, we first identified LTR elements using LTRharvest from the GenomeTools package v1.6.5[92]. The paired LTR sequences for each retrotransposon (Supplementary Data 9) were aligned with MEGA X[69] and genetic distances were calculated using Kimura's two-parameter model[75]. To estimate the timing of the inversion in the M haplotype relative to the F haplotype in *M. rotundifolia*, we identified the boundaries of the inversion by aligning the M haplotype of Trayshed and DVIT1750 to their corresponding F haplotypes using NUCmer from MUMmer v4.0.0 with the "-mum" option[80]. Coordinates of the inversion ends are reported in Supplementary Table 2, and the associated sequences are provided in Supplementary Data 10. Genetic distances were assessed using Kimura's two-parameter model[75] in MEGAX[69]. Divergence times were calculated as below:

$$T = K/2\mu \times \text{generation time} \qquad (1)$$

where $K$ is the genetic distance and $\mu$ is the mutation rate. A generation time of 3 years and a nucleotide substitution rate of $2.5 \times 10^{-9}$ substitutions per base per year were assumed[5].

## Transcription factor-binding site analysis

For each haplotype, promoter sequences were extracted with a maximum of 3 kbp upstream regions from the gene transcriptional start sites. TF-binding sites were identified using the R packages TFBSTools v.1.38[93] and the JASPAR2018 v.1.1.1[94].

## Gene expression analysis

Samples were randomly subsampled to 25 million reads with seqtk sample from the package seqtk v.1.2-r101-dirty (https://github.com/lh3/seqtk) and the parameter -s100 before trimming. RNA-seq reads were trimmed using Trimmomatic v.0.36[82] and the following settings: "LEADING:3 TRAILING:3 SLIDINGWINDOW:10:20 MINLEN:36". Transcript abundance was assessed using Salmon v.1.5.1[95] and these parameters: "-gcBias -seqBias -validateMappings". For each accession, a transcriptome index file was built using Trayshed protein-coding sequences ver2.2 and the Trayshed genome v.2.1[29] as decoy, and a k-mer size of 31. Counts were imported using the R package tximport v.1.20.0[96] and combined at gene-level for genes with alternative transcripts. Statistics about the RNA-seq analysis can be retrieved in Supplementary Data 11. Because of a low percentage of reads aligning on Trayshed protein-coding sequences, the third bioreplicate of ovaries from DVIT1750 was removed from the dataset. Gene expression of the two alleles of *VviYABBY3* represented in Fig. 6b corresponds to the average of the $\log_2$-transformed (TPM + 1) detected in the three bioreplicates.

## Reporting summary

Further information on research design is available in the Nature Portfolio Reporting Summary linked to this article.

## Data availability

Sequencing data generated in this study have been deposited at NCBI under the BioProject PRJNA1151724. Genome sequences and gene annotation files have been deposited at Zenodo [https://zenodo.org/records/13362874]. Source data are provided with this paper.

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

## Acknowledgements

We would like to thank Bernard Prins and Claire Heinitz (National Clonal Germplasm, USDA-ARS) for providing information about the Vitaceae material, Ernesto Sandoval (University of California Davis) and Daniel Pfarr (Sacramento State University) for providing leaf material from *L. coccinea* 1464, and Malin Petersen (University of British Columbia, Canada) for her help during the plant tissues collection. This work was funded by the NSF grant #1741627, USDA NIFA Award # 2022-51181-38240, and partially supported by the Ray Rossi Endowment and the E. & J. Gallo Winery.

## Author contributions

M.M. and D.C. designed the project. D.C. secured the funds and supervised the project. M.M. and J.P.L. collected the plant material. R.F.-B. extracted DNA and RNA, and prepared sequencing libraries. M.M. and A.M. assembled the genomes. M.M., N.C, A.M., and V.R. performed the data analyses. M.M. and D.C. wrote the manuscript.

## Competing interests

The authors declare no competing interests.
