## [Peer Review file · Nature Communications]

Evolutionary conservation of the grape sex-determining region in angiosperms and emergence of dioecy in Vitaceae

Corresponding Author: Professor Dario Cantu

Version 0:

Reviewer comments:

Reviewer #1

(Remarks to the Author)

The manuscript 'Evolutionary conservation of the grape sex-determining region in angiosperms and emergence of dioecy in Vitaceae' by Mélanie Massonnet et al. explored the evolutionary conservation of the Vitis SDR across angiosperms and Vitaceae family, and through comparative analysis, they inferred that different SDRs may exist in Tetrastigma, while muscadine grapes have SDRs similar to those in Vitis. They concluded that a locus that is highly conserved in flowering plants evolved into the SDRs in the common ancestor of the two genera of grapes. This conclusion is interesting at first glance, but upon closer examination it is mostly descriptive, predictable, and partially published, with my most important questions being the following:

1. First, and most importantly, as reported in a previous study (<https://doi.org/10.1038/s41467-020-16700-z>), Vitis and muscadine had similar male (M) and female (F) haplotypes, and the overall structure, gene content and order, and even mutations were clustered by sex among these two lineages, which strongly supported that their SDRs originated from their common ancestor. Therefore, the repetition of this result in this manuscript lacks novelty.

2. The authors used 42 plant genomes to assess the conservation of grape SDR regions. The authors should first state how they dealt with the impact of frequent whole genome duplications in the plant kingdom on this analysis. Should only one of the paralogous copies after whole genome duplication maintain collinearity be considered conserved, or should multiple copies maintain collinearity? How much do different strategies affect the conclusions? What I want to know more is how many collinear blocks with similar conservation to grape SDR were found by the authors in so many species, and what functionally related genes are there? Another issue is that this genomic region, which is conserved among angiosperms, is not uncommon to see higher conservation in the Vitaceae family, which has a smaller divergence scale. So what is the hypothesis that the authors set out to test in this article? That the sex-determining regions of dioecious plants should originate in regions of the genome that are less conserved during evolution? So what is the authors' hypothesis? Should the sex-determining regions of dioecious plants originate from regions of the genome that are less conserved during evolution, or vice versa? Or is the one described in this article just a special case? How does it differ from other SDRs in other dioecious plants?

3. As far as I know, the sex system and SDR of *Tetrastigma voinieranum* have not been studied yet. So the authors should state what is the sex of the sequenced individual of this species, male or female? And do they know the sex system of this species? Similarly, the authors claim that they aligned the short sequencing reads of 13 *Tetrastigma* species (by the way, I found more species in the supplementary data) to the reference genome they assembled. Whether these resequencing data were generated in this article or downloaded from other database, the reads number of the sequencing data, the alignment rate, genome coverage, and the sex of these sequenced individuals should be explained in the article and the corresponding tables, so as to better evaluate their results in the linkage analysis of the VSR of *Tetrastigma* spp.. In addition, I found that the authors used polymorphism data from different species to assess the degree of linkage disequilibrium, which seems difficult to understand in population genetics. This is because most polymorphic sites are not shared between species. Will this affect the conclusion of this article? I hope the authors can give an assessment. For similar reasons, I also question the authors' comparative analysis of heterozygosity, which is more susceptible. Therefore, I feel that the authors' conclusion that VSR in *T. voinieranum* is not related to sex is not supported by very solid evidence.

4. About the sex-determining region of *Muscadinia rotundifolia*, I found that the Fry (female) and DVIT1750 (male) genomes

assembled by the authors are much smaller than the previous Trayshed. The authors should evaluate whether these genomes are fully assembled and how many gaps in them, especially in SDRs. The results of the structural comparison of the four F haplotypes and the two M haplotypes, as well as the expression of related genes, did not change much from the previous conclusions. Therefore, the authors should point out the highlights and novelty of their research results.

Reviewer #2

(Remarks to the Author)

The manuscript represents a significant piece of work that explores grape sex-determining region (SDR) evolution in angiosperms broadly and in the grape family in particular. The genomic basis of sex determining is fundamental for the grape industry and for the evolutionary biology community. The methods and analyses are sound. I only suggest some additional sampling in angiosperms and in the grape family to further strengthen the study. Other suggested changes are minor.

(1) The first part of the paper will benefit from inclusion of several major early diverged angiosperm lineages such as *Amborella trichopoda* (the single living representative of the sister lineage to all other extant flowering plants), *Aristolochia fimbriata* (magnoliid), *Ceratophyllum demersum* and *Chloranthus sessilifolius* beyond mostly Rosids, Asterids and monocots sampled in the study to assess evolutionary conservation of grape sex-determining region in angiosperms. The comparative results were largely based on sampling rosids, asterids and some monocots, leaving out the early diverged lineages in the ANA grade and the Magnoliids. The paper also treated Nymphaeales and Ranunculales as belonging to monocots. They are not. With the suggested expansion of the sampling, the authors may properly assess the evolution of SDR across angiosperms, with the ANA grade and Magnoliids included.

(2) This manuscript used *Muscadinia* as a distinct genus closely related to *Vitis*. The *Muscadinia* lineage is often not recognized as a genus, instead, it has been recognized as a subgenus of *Vitis*, i.e., *Vitis* subgenus *Muscadinia*. The genus *Vitis* is recognized as a genus that includes the *Muscadinia* lineage, on the basis of two synapomorphic characters: dioecy and calyptrate petals. The analyses of the paper also support the origin of the dioecy trait in Viteae before the divergence of the *Vitis* subgenus *Vitis* and subgenus *Muscadenia*.

(3) For the divergence of Viteae (*Vitis* and its allies), the authors may also check and cite Nie et al. Climate-influenced boreotropical survival and rampant introgressions explain the thriving of New World grapes in the north temperate zone. *J Integr Plant Biol.*, 2023, 65(5): 1183-1203.

(4) For SDR conservation with Vitaceae, I suggest the authors include the *Cissus rotundifolia* genome, which was published by Xin et al. 2022. The inclusion will expand the coverage of the phylogenetic diversity of *Cissus*, the largest genus of Vitaceae. Your current sampling includes two species both from the Neotropics. *Cissus rotundifolia* is from Africa, a region rich in *Cissus* (also the ancestral area of *Cissus*).

(5) *Tetrastigma* is not really from tropical southern hemisphere areas (see page 15, line 417). *Tetrastigma* is widely distributed in subtropical and tropical Asia to Australia. So make revisions accordingly.

(6) P. 16, line457, change “Vitacea” to “Vitaceae”.

(7) Supplementary Fig. 9, change “*V. sylvestris* ssp. *sylvestris*” to “*V. vinifera* ssp. *sylvestris*.” By the way, ssp. should not be italicized.

(8) Vitaceae in the title and in many parts of the manuscript should not be italicized.

(9) Page 6, line 116, add “ancient” before “tracheophyte Selaginella...”.

(10) Page 6, line 116, change “embryophyte” to “bryophyte”. Embryophyte will be too broad for this context. You really mean bryophyte for *Physcomitrium*, at least it's so much more precise systematically.

(11) *Ampelopsis glandulosa* var. *brevipedunculata*, change to: *Ampelopsis glandulosa* var. *brevipedunculata*. The variety name *brevipedunculata* needs to be italicized. Also be sure this change is made in supplemental material, e.g., Supplemental Data 3.

(12) Change *Cayratia japonica* in Supplemental Data 3 - *Causonis japonica*, to reflect the generic changes concerning *Cayratia*, see Wen et al. 2018, and Parmar et al. 2021 [Phylogeny, character evolution and taxonomic revision of *Causonis*, a segregate genus from *Cayratia* (Vitaceae), *Taxon* ..]. You actually used the correct names in Supplemental Data 6.

Reviewer #3

(Remarks to the Author)

This manuscript presents a comparative analysis of the genetic sex-determining region in the grapevine *Vitis sylvestris* among Angiosperms. The first part of the manuscript consists of a broad comparative analysis of this region among angiosperms, showing that the region is conserved in flowering plants, but not in outgroups. The authors report that this is

the first conserved region specialized in flowering that has been shown to be involved in sex determination. While the results on the conservation of this region are robust, it is unclear what this implies for sex chromosome formation in plants, since all other known cases of sex-determining regions do not come from conserved regions.

In a second part, the authors study this sex-determining region in muscadine grapevines (*Vitis rotundifolia*) and in the genus *Tetragymna* of the family Vitaceae, both models having separate sexes. Based on genome assemblies, resequencing, polymorphism analysis and linkage disequilibrium, they find that the genus *Tetragymna* does not have the same sex-determining region that *V. sylvestris*. Of course, their data certainly allow a genome-wide approach that would allow the search for the sex-determining region in *Tetragymna*, but I imagine that will be done in a future study.

Concerning *Vitis rotundifolia*, the authors find that the same sex-determining region in *Vitis sylvestris* is most likely responsible for genetic sex. However, while I agree with their analyses, I disagree with their main conclusion, which is a unique origin of dioecy in the genus *Vitis*. They find that the 8-base pair deletion mutation present in INP1, the male candidate, is shared between the two *Vitis* species. This mutation therefore appeared prior to the separation between *V. sylvestris* and *V. rotundifolia*. However their own analyses contradict a single origin of dioecy in *V. sylvestris* and *V. rotundifolia*. Here is the list of arguments against this conclusion :

- they find that there is an inversion between the female haplotype and the male haplotype in *V. rotundifolia* while there is none in *V. sylvestris*. This suggests a different mechanism of recombination suppression in the two species.
- they find that the age of the recombination suppression, inferred from the synonymous polymorphism rate (dS) differs between *V. sylvestris* and *V. rotundifolia* (older in *V. rotundifolia*).
- they find that no gene except INP1 (responsible for male sterility) and the flanking gene Exostosin shows trans-specific polymorphism, which should be the case if the recombination suppression was prior to the separation.

In fact, their only argument for a common origin of dioecy (apart from the antiquity of male INP1) is the presence of sex-linked mutations in the promoter of YABBI3, the other candidate gene in the sex-determining region. But these results are only shown as a neighbor-joining tree, we would like to see the sequence alignment to see the raw results since it is so important.

I find that this study rather suggests that there were two independent recombination suppression events in *V. sylvestris* and *V. rotundifolia* that created these sex-determining regions, at the level of a region that had a genomic predisposition with a male-sterility mutation.

So I think the manuscript would be clearer if the authors reformulated their conclusions to be more in line with their own analyses, and showed more in-depth analysis of the YABBI3 promoter.

Version 1:

Reviewer comments:

Reviewer #1

(Remarks to the Author)

In the revised version of the manuscript "Evolutionary conservation of the grape sex-determining region in angiosperms and emergence of dioecy in Vitaceae", the authors have addressed several of my previous concerns, particularly regarding the novelty of this study compared to their 2020 publication. While their responses are generally satisfactory, I have a few minor suggestions for further improvement:

Conservation of the *Vitis* SDR and Flanking Regions: The authors state that the *Vitis* SDR consists of 12–14 protein-coding genes and used these genes to assess conservation across angiosperms. I recommend that the authors include a figure illustrating the collinearity results for these genes. Additionally, it would be valuable to investigate whether the genes flanking the SDR also exhibit collinearity and to what extent this conservation extends. Another important point is that the authors' evaluation of the functional significance of SDR conservation is not clear. The article found that grape SDR is highly conserved in angiosperms, but the significance and specific mechanism of this conservation in sex determination are difficult to explain. In the phylogenetic analysis, the authors expanded the gene set to 21, suggesting that the conservation might extend beyond the SDR. If so, the functional relevance of these flanking genes may not be limited to flower development, and this broader genomic context should be explored.

Estimation of Divergence Times: In the Results section, the authors inferred the timing of inversion events and haplotype differentiation based on 'genetic distances'. However, they did not specify the exact genetic distances used, the sequences analyzed, or the reference points for calculating divergence times and their confidence intervals. This lack of detail could lead to misinterpretations regarding the origin of the SDR, the timing of recombination suppression, and the mutation rates in coding regions and repetitive sequences. Clarifying these methodological details is crucial to ensure the robustness of the conclusions.

Analysis of *Tetragymna* Species: I remain cautious about the analysis of *Tetragymna* species. The authors performed linkage disequilibrium (LD) analysis by pooling data from 13 different species, but the distribution of SNP frequencies (which are likely species-specific) and the sequence divergence between species (which may not be related to sex linkage) could significantly bias the results. Therefore, the authors should exercise caution when interpreting these findings and avoid

overgeneralizing the conclusions. It would be beneficial to provide a more detailed analysis of SNP frequencies and species-specific variations to validate the LD results.

The authors found 15 distant SNP sites on the promoter of VviYABBY3, one of which is a binding site for the MYB transcription factor, and speculated that its variation between M and F haplotypes may affect its expression level. The authors should verify the regulatory and activation capabilities of this hypothesis, otherwise it is difficult to believe that the difference in expression of this gene between male and female organs is determined by the mutation of this site.

Reviewer #4

(Remarks to the Author)

Reviewer #5

(Remarks to the Author)

Overall, the authors appear to have responded appropriately to the comments from Reviewer 3. However, one concern remains regarding the initial comment:

“While the results on the conservation of this region are robust, it is unclear what this implies for sex chromosome formation in plants, since all other known cases of sex-determining regions do not come from conserved regions.”

What Reviewer 3 is essentially asking here is what the relatively unchanged nature of the sex-determining region (SDR) in Vitaceae implies for the broader understanding of sex chromosome evolution in plants. This question arises because in most other dioecious plant species, SDRs tend to evolve rapidly, often undergoing large-scale degeneration (on the order of Mbp) and evolving independently. Consequently, the concept of conservation from an ancestral region is far from those cases. In this light, although the authors' manuscript demonstrates that the SDR in Vitaceae has been conserved from its ancestral state (and such case would be the first observed in the field of plant sex determination), it does not clearly address what this finding means in the context of previously known patterns of sex chromosome evolution.

In the latter part of the authors' response, the authors refer to dS values, which is indeed relevant. However, the dS results suggest that, despite evolutionary time having passed, the changes in genes located in the SDR are minimal, and unlike in other plant species, a male-hemizygous state (if in a XY system) has not developed. This may represent a rare or exceptional case. Reviewer 3 may be implying that this unique feature of conservation could allow for a novel approach to sex chromosome analysis.

Furthermore, there remain several misleading expressions in the revised manuscript. Some phrases could be misinterpreted to suggest that conservation exists between SDRs across different species, which likely reflects a conceptual gap between Reviewer 3 and the authors. I agree that the SDR in Vitaceae quite conserves the original genomic state, albeit not SDRs in other dioecious species. These should be revised for proper understanding. In reality, plant SDRs have evolved convergently from different genes and regions. If any "conservation" exists, it may only involve gene function (but note that, in the family Salicaceae or the genus Actinidia, SDRs undergo frequent turnover, which casts doubt on the conservation of even gene function across lineages).

In connection with Reviewer 3's comments, the following points should also be reconsidered:

Line 484: Use of the term "sexual dimorphism" (or "sexually antagonistic trait") is inappropriate. This term refers to traits that confer an advantage to only either of the two sexes, male or female (unrelated to sex determination itself), and there is no discussion of sexual dimorphism in this manuscript. The intended meaning here appears to be different from how the term is used.

Estimation of evolutionary timescales based on dS values: The interpretation presented in Line 663 seems to overestimate the timing of recombination suppression or species divergence. It may be possible that the nucleotide substitution rate was calculated "per generation" rather than "per year", which would significantly alter the conclusions regarding evolutionary timing.

Version 2:

Reviewer comments:

Reviewer #1

(Remarks to the Author)

The authors have addressed all my major concerns satisfactorily, and I have no further comments at this stage.

Reviewer #5

(Remarks to the Author)

The authors have provided appropriate responses to my queries. I have no further comments, and I look forward to the publication.

Reviewer #1 (Remarks to the Author):

The manuscript ‘Evolutionary conservation of the grape sex-determining region in angiosperms and emergence of dioecy in Vitaceae’ by Mélanie Massonnet et al. explored the evolutionary conservation of the *Vitis* SDR across angiosperms and Vitaceae family, and through comparative analysis, they inferred that different SDRs may exist in Tetrastigma, while muscadine grapes have SDRs similar to those in *Vitis*. They concluded that a locus that is highly conserved in flowering plants evolved into the SDRs in the common ancestor of the two genera of grapes. This conclusion is interesting at first glance, but upon closer examination it is mostly descriptive, predictable, and partially published, with my most important questions being the following:

1. First, and most importantly, as reported in a previous study (<https://doi.org/10.1038/s41467-020-16700-z>), *Vitis* and muscadine had similar male (M) and female (F) haplotypes, and the overall structure, gene content and order, and even mutations were clustered by sex among these two lineages, which strongly supported that their SDRs originated from their common ancestor. Therefore, the repetition of this result in this manuscript lacks novelty.

While our previous study (Massonnet et al., 2020) provided initial insights into the structural similarities between the *Vitis* SDR and the muscadine SDR in *M. rotundifolia* Trayshed, it left key questions unanswered. The precise boundaries of the muscadine SDR remained undefined, and our analysis was restricted to a single male genome, limiting our ability to investigate sequence clustering by flower sex type. In this study, we address these gaps by generating and analyzing phased SDR haplotypes from both male and female muscadine individuals, enabling us to (1) define the muscadine SDR more precisely using linkage disequilibrium analysis, (2) identify a homozygous deletion in *VviINPI* as a candidate male-determining mutation, (3) confirm that the SDR inversion previously observed in Trayshed is conserved in the muscadine M haplotype, and (4) expand promoter analysis and gene expression profiling beyond *Vitis* to muscadine, providing new evidence on the potential role of *VviYABBY3* in female sterility. These findings go beyond our previous study and provide novel insights into the evolution and function of the SDR in muscadines.

We highlighted these points in the following sentences:

Page 3, “A high-density linkage map has located the SDR in *M. rotundifolia* on the same chromosomal region as in *Vitis* spp. (Lewter et al., 2019), suggesting that both genera have evolved from a common dioecious ancestor. However, the precise boundaries of the muscadine SDR have yet to be defined, making it unclear whether the same genes regulate flower sex in both grape genera. A previous study identified the same candidate male-sterility mutation in the female-associated (F) haplotype of the male *M. rotundifolia* cv. Trayshed as in *Vitis* species (Massonnet et al., 2020). However, for the candidate female-suppressing gene in *Vitis* spp., *VviYABBY3*, Trayshed lacked the two non-synonymous SNPs specific to the male-associated (M) allele in *Vitis*, and the gene expression of *VviYABBY3* in muscadine flowers has yet to be investigated.”

Page 7: “In a previous study, we determined the gene content of the two VSR haplotypes of the male *M. rotundifolia* Trayshed (Massonnet et al., 2020). However, the boundaries of the SDR in *M. rotundifolia*, and its candidate sex-determining genes are still unknown.”

Page 11: “In this study, we aimed to better characterize the muscadine SDR to assess the commonalities and differences between the two grape SDRs.”

Page 9: “In summary, defining the boundaries of the *M. rotundifolia* SDR revealed that the muscadine and *Vitis* SDRs share similar boundaries. The generation of additional *M. rotundifolia* genomes confirmed the presence of an inversion in the M haplotype relative to the F haplotype. It also demonstrated that the 8-bp deletion in *VviINP1* is homozygous in female muscadines, suggesting that *VviINP1* could also function as the male-determining gene in muscadines. Sequence analysis of SDR genes confirmed the absence of an M-specific mutation shared between *Vitis* and *M. rotundifolia*, whereas M-specific mutations affecting transcription factor binding sites were identified in the promoter region of *VviYABBY3*. As in *Vitis* spp., the M allele of *VviYABBY3* was more highly expressed in the ovaries of male flowers compared to the F allele, and the F allele in male flowers exhibited lower expression than in female flowers. This suggests that the M allele of *VviYABBY3* may also function as the female-suppressing factor in muscadines. Altogether, our results indicate that the *Vitis* and *M. rotundifolia* SDRs not only share similar boundaries but also harbor similar candidate sex-determining genes, suggesting a common evolutionary origin”.

2. The authors used 42 plant genomes to assess the conservation of grape SDR regions. The authors should first state how they dealt with the impact of frequent whole genome duplications in the plant kingdom on this analysis. Should only one of the paralogous copies after whole genome duplication maintain collinearity be considered conserved, or should multiple copies maintain collinearity? How much do different strategies affect the conclusions? What I want to know more is how many collinear blocks with similar conservation to grape SDR were found by the authors in so many species, and what functionally related genes are there?

We recognize that whole-genome duplications (WGDs) can complicate assessments of homologous regions by generating multiple paralogous copies. To account for this, we conducted a comprehensive collinearity analysis using GENESPACE (Lovell et al., 2022), which enabled us to identify all homologous regions, including both orthologous and paralogous copies. Across a 12-gene window, we detected an average of ~ 1.1 to 3.0 ± 1.1 homologous regions, with their distribution detailed in Supplementary Fig. 2.

Specifically, for the *Vitis* SDR, GENESPACE identified one to four homologous regions across the analyzed angiosperms. Notably, species such as *Actinidia chinensis*, *Glycine max*, *Arabidopsis thaliana*, and *Daucus carota* ssp. *sativus*, which have undergone at least two WGDs since the ancient γ hexaploidization event, retained four homologous SDR-like regions (Supplementary Fig. 3). Despite these duplications, our analyses revealed that the inclusion of both orthologous and paralogous regions did not substantially alter our conclusions: the number of collinear genes and their rank-normalized gene conservation scores per 12-gene window remained consistent. This supports the interpretation that the genomic region corresponding to the *Vitis* SDR is highly conserved across angiosperms.

Page 5: Because several plant species used in the gene collinearity analysis had undergone whole-genome duplication(s) (Clark and Donoghue, 2018), which could potentially affect the analysis of conservation of the *Vitis* SDR, we investigated whether gene collinearity was maintained in all homologous (*i.e.*, orthologous and paralogous) regions by repeating the analysis using GENESPACE (Lovell et al., 2022). No synteny was detected between the haplotype 1 of Cabernet Sauvignon and the genome of the thirteen non-flowering plants. In contrast, among angiosperms, we identified at least one homologous region for an average of $1,755.9 \pm 347.5$ 12-gene windows. The number of homologous regions per collinear 12-gene window ranged from approximately 1.1 in *Cannabis sativa* and in each haplotype of *Fragaria* species, to 3.0 ± 1.1 in *Actinidia chinensis* and *Glycine max* (Supplementary Fig. 2). Moreover, four homologous regions corresponding to the *Vitis* SDR were found in *Actinidia chinensis* and *Glycine max*, *Arabidopsis thaliana* and *Daucus carota* ssp. *sativus* (Supplementary Fig. 3). All these plant species had experienced two whole-genome duplications since the ancient hexaploidization event γ (Jaillon et al., 2007; Huang et al., 2013; Clark and Donoghue, 2018). In terms of number of collinear genes and their rank-normalized gene conservation score per 12-gene window, similar results were found when including both orthologous and paralogous regions, although the difference between the *Vitis* SDR and the other 19 grape chromosome was slightly reduced, but still significant (P value $< 1 \times 10^{-9}$; Fig. 2c-d). These results support that the genomic region corresponding to the *Vitis* SDR is highly conserved among angiosperms”.

Another issue is that this genomic region, which is conserved among angiosperms, is not uncommon to see higher conservation in the Vitaceae family, which has a smaller divergence scale. So what is the hypothesis that the authors set out to test in this article? That the sex-determining regions of dioecious plants should originate in regions of the genome that are less conserved during evolution? So what is the authors' hypothesis? Should the sex-determining regions of dioecious plants originate from regions of the genome that are less conserved during evolution, or vice versa? Or is the one described in this article just a special case? How does it differ from other SDRs in other dioecious plants?

We appreciate the reviewer’s question and acknowledge that Vitaceae were not included in the genome-wide collinearity analysis to assess locus conservation. Instead, within Vitaceae, our analysis focused on the structure and gene content of the SDR-like region at a higher resolution, given the smaller divergence scale and our primary objective of investigating the evolution of this locus within the Vitaceae family.

To our knowledge, this study is the first evolutionary analysis of a plant SDR across distantly related species. Our findings indicate that the *Vitis* SDR is embedded within a highly conserved flowering-related genomic region. This observation is particularly intriguing, as it suggests that dioecy in *Vitis* may have evolved through modifications within an already conserved genomic framework involved in flower development. In this scenario, genes originally regulating broader aspects of flowering may have been co-opted for sex determination through key regulatory changes.

However, we do not suggest that SDRs must always arise in conserved genomic regions. Rather, our study highlights one possible evolutionary trajectory, where a functionally relevant, evolutionarily stable genomic region was repurposed for sex determination. This contrasts with other dioecious plants, where SDRs have frequently been associated with rapidly evolving or less conserved genomic regions. Whether the *Vitis* SDR represents a rare exception or part of a broader

evolutionary pattern remains an open question. Future comparative analyses across diverse dioecious lineages will be essential to determine whether conserved genomic regions commonly serve as the foundation for SDR formation in plants.

This point is discussed page 10: “The conservation of other plant SDRs and their role in flowering processes remain open areas for exploration. Evaluating the conservation of SDRs in distantly related plant species would help determine whether SDRs originate in highly conserved genomic regions across plants or if the *Vitis* SDR represents a unique case”.

3. As far as I know, the sex system and SDR of *Tetrastigma voinieranum* have not been studied yet. So the authors should state what is the sex of the sequenced individual of this species, male or female? And do they know the sex system of this species?

The sex determination system in *Tetrastigma* has not yet been characterized, and the flower sex of the individual used in our analysis remains unknown. However, this uncertainty does not affect our LD analysis, as LD signals are expected to be strong in a dioecious species regardless of the specific sex of the sampled individual.

That said, we acknowledge the limitations of our LD analysis due to low genome coverage, and we recognize that additional information on the flower sex type of *Tetrastigma* species would be valuable for identifying the precise location and characteristics of the SDR in this genus. We have now explicitly discussed these points in the manuscript.

Page 6: “Given that *Tetrastigma* spp. are dioecious with an unknown sex determination system, and two VSRs were found in the genome of *T. voinieranum*, we investigated whether the VSR could also be associated with sex determination in *Tetrastigma*. To test this, DNA-seq short reads from thirteen *Tetrastigma* species were retrieved from NCBI and aligned onto the haplotype 1 of *T. voinieranum* (Supplementary Data 6)”

Page 11: “Low LD indicates that the region is unlikely associated with sex determination, suggesting that dioecy emerged in two distinct regions in the Cayratieae and Viteae clades. However, the genome coverage of the *Tetrastigma* species retrieved from NCBI was low (< 3.9X; Supplementary Data 6). Despite selecting species with at least 1.9X coverage for variant calling, a low number of sites with genotyping information was detected (3,623 sites across 902.5 kbp). The low genome coverage of the samples (<3.9×), the quality of the alignments, or the genetic distance between species may have posed limitations for variant calling and, consequently, the LD analysis. To address this, sequencing additional *Tetrastigma* species and varieties with higher sequencing coverage would help validate the absence of linkage constraints in the region. When combined with flower sex type phenotyping, this approach could facilitate the identification of the *Tetrastigma* SDR”.

Similarly, the authors claim that they aligned the short sequencing reads of 13 *Tetrastigma* species (by the way, I found more species in the supplementary data) to the reference genome they assembled. Whether these resequencing data were generated in this article or downloaded from other database, the reads number of the sequencing data, the alignment rate, genome coverage, and the sex of these sequenced individuals should be explained in the article and the corresponding tables, so as to better evaluate their results in the linkage analysis of the VSR of *Tetrastigma* spp..

Information about the origin of the sequencing data, the number of reads, number of high-quality reads, alignment rate, genome coverage, and known flower sex type were added to the Supplementary Data 6. Samples with at least 1.9X coverage were used for the variant calling analysis. This information was added to the manuscript.

Page 15: “For *Tetrastigma*, samples retrieved from NCBI with at least 1.9X coverage were used for variant calling.”

The limitation of a low genome coverage is discussed page 11: “However, the genome coverage of the *Tetrastigma* species retrieved from NCBI was low (< 3.9X; Supplementary Data 6). Despite selecting species with at least 1.9X coverage for variant calling, a low number of sites with genotyping information was detected (3,623 sites across 902.5 kbp). The low genome coverage of the samples (<3.9×), the quality of the alignments, or the genetic distance between species may have posed limitations for variant calling and, consequently, the LD analysis. To address this, sequencing additional *Tetrastigma* species and varieties with higher sequencing coverage would help validate the absence of linkage constraints in the region. When combined with flower sex type phenotyping, this approach could facilitate the identification of the *Tetrastigma* SDR”.

In addition, I found that the authors used polymorphism data from different species to assess the degree of linkage disequilibrium, which seems difficult to understand in population genetics. This is because most polymorphic sites are not shared between species. Will this affect the conclusion of this article? I hope the authors can give an assessment. For similar reasons, I also question the authors' comparative analysis of heterozygosity, which is more susceptible. Therefore, I feel that the authors' conclusion that VSR in *T. voinieranum* is not related to sex is not supported by very solid evidence.

Linkage disequilibrium (LD) analysis has previously been applied across multiple *Vitis* species to define the boundaries of the *Vitis* SDR (Zou et al., 2021), suggesting that a similar approach could be used for *Tetrastigma*. However, we acknowledge the limitations of our current analysis and the concerns raised by the reviewer. Specifically, our analysis identified only 3,623 sites with genotyping information within the VSR of *T. voinieranum*, while 84,809 bases were covered by at least one sequencing read (9.4% of the VSR). This relatively low proportion suggests that factors such as low genome coverage (<3.9X), alignment quality, and genetic divergence between species may have affected the accuracy of variant calling and, consequently, our LD analysis. Regarding the comparative analysis of heterozygosity, we agree that heterozygosity levels can be influenced by multiple factors, including sequencing depth and genetic diversity within and between species. Given these limitations, we acknowledge that our conclusion regarding the lack of a sex-related SDR in *Tetrastigma* should be interpreted with caution.

This point was added in page 7: “Concerning the VSR, 3,623 sites with genotyping information were identified across the thirteen species of *Tetrastigma*, with each species averaging 448.1 ± 233.7 SNPs. The number of sites with genotyping information is very low considering the region size (902.5 kbp). However, a total of 84,809 bases were covered by at least one sequencing read in all thirteen *Tetrastigma* species (9.4% of the VSR). This suggests that either the low genome coverage of the samples (< 3.9X), the quality of the alignments, or the genetic distance between species, might have limited the variant calling”.

Page 7: “Taken together, these results suggest that the VSR in *T. voinieranum* is probably not associated with sex determination”.

Page 11: “Low LD indicates that the region is unlikely associated with sex determination, suggesting that dioecy emerged in two distinct regions in the Cayratieae and Viteae clades. However, the genome coverage of the *Tetrastigma* species retrieved from NCBI was low (< 3.9X; Supplementary Data 6). Despite selecting species with at least 1.9X coverage for variant calling, a low number of sites with genotyping information was detected (3,623 sites across 902.5 kbp). The low genome coverage of the samples (<3.9×), the quality of the alignments, or the genetic distance between species may have posed limitations for variant calling and, consequently, the LD analysis. To address this, sequencing additional *Tetrastigma* species and varieties with higher sequencing coverage would help validate the absence of linkage constraints in the region. When combined with flower sex type phenotyping, this approach could facilitate the identification of the *Tetrastigma* SDR”.

4. About the sex-determining region of *Muscadinia rotundifolia*, I found that the Fry (female) and DVIT1750 (male) genomes assembled by the authors are much smaller than the previous Trayshed. The authors should evaluate whether these genomes are fully assembled and how many gaps in them, especially in SDRs.

The genomes of Fry (female) and DVIT1750 (male) were sequenced using PacBio CLR reads and assembled with FALCON-Unzip, resulting in partially phased genomes. Variations in genome size between assemblies—such as the smaller sizes of Fry and DVIT1750 compared to Trayshed—can be attributed to differences in heterozygosity levels between haplotypes, which influence the phasing and completeness of assembled sequences.

Importantly, the sizes of the Fry and DVIT1750 assemblies are consistent with other grape genomes generated using the same sequencing technology and assembly approach (Massonnet et al., 2020). Unlike Trayshed, these genomes were not scaffolded, meaning they do not contain gaps introduced by scaffolding procedures.

To specifically address concerns about the SDR, we confirmed that all six muscadine SDR haplotypes were completely assembled without gaps using FALCON-Unzip, ensuring that the SDR sequences used in our analysis are fully resolved.

The results of the structural comparison of the four F haplotypes and the two M haplotypes, as well as the expression of related genes, did not change much from the previous conclusions. Therefore, the authors should point out the highlights and novelty of their research results.

The highlights of the new results were added in the manuscript.

Page 9: “In summary, defining the boundaries of the *M. rotundifolia* SDR revealed that the muscadine and *Vitis* SDRs share similar boundaries. The generation of additional *M. rotundifolia* genomes confirmed the presence of an inversion in the M haplotype relative to the F haplotype. It also demonstrated that the 8-bp deletion in *VviINP1* is homozygous in female muscadines, suggesting that *VviINP1* could also function as the male-determining gene in muscadines. Sequence analysis of SDR genes confirmed the absence of an M-specific mutation shared between *Vitis* and *M. rotundifolia*, whereas M-specific mutations affecting transcription

factor binding sites were identified in the promoter region of *VviYABBY3*. As in *Vitis* spp., the M allele of *VviYABBY3* was more highly expressed in the ovaries of male flowers compared to the F allele, and the F allele in male flowers exhibited lower expression than in female flowers. This suggests that the M allele of *VviYABBY3* may also function as the female-suppressing factor in muscadines. Altogether, our results indicate that the *Vitis* and *M. rotundifolia* SDRs not only share similar boundaries but also harbor similar candidate sex-determining genes, suggesting a common evolutionary origin”.

Reviewer #2 (Remarks to the Author):

The manuscript represents a significant piece of work that explores grape sex-determining region (SDR) evolution in angiosperms broadly and in the grape family in particular. The genomic basis of sex determining is fundamental for the grape industry and for the evolutionary biology community. The methods and analyses are sound. I only suggest some additional sampling in angiosperms and in the grape family to further strengthen the study. Other suggested changes are minor.

(1) The first part of the paper will benefit from inclusion of several major early diverged angiosperm lineages such as *Amborella trichopoda* (the single living representative of the sister lineage to all other extant flowering plants), *Aristolochia fimbriata* (magnoliid), *Ceratophyllum demersum* and *Chloranthus sessilifolius* beyond mostly Rosids, Asterids and monocots sampled in the study to assess evolutionary conservation of grape sex-determining region in angiosperms. The comparative results were largely based on sampling rosids, asterids and some monocots, leaving out the early diverged lineages in the ANA grade and the Magnoliids. The paper also treated Nymphaeales and Ranunculales as belonging to monocots. They are not. With the suggested expansion of the sampling, the authors may properly assess the evolution of SDR across angiosperms, with the ANA grade and Magnoliids included.

Thank you very much for the suggestion. The four genomes were added to the analysis.

(2) This manuscript used *Muscadinia* as a distinct genus closely related to *Vitis*. The *Muscadinia* lineage is often not recognized as a genus, instead, it has been recognized as a subgenus of *Vitis*, i.e., *Vitis* subgenus *Muscadinia*. The genus *Vitis* is recognized as a genus that includes the *Muscadinia* lineage, on the basis of two synapomorphic characters: dioecy and calyprate petals. The analyses of the paper also support the origin of the dioecy trait in Viteae before the divergence of the *Vitis* subgenus *Vitis* and subgenus *Muscadenia*.

We understand the reviewer’s point of view. The fact that *Vitis* spp. and *M. rotundifolia* Small are from two separate genera has been a long-standing controversy. Several morphological, anatomical, and genetic differences support the distinction of *Muscadinia* as a separate genus. For instance, muscadine grapes (*M. rotundifolia* Small) are characterized by fruit borne in many clusters with few berries per cluster, whereas bunch grapes (*Vitis* spp.) produce many berries per fruit cluster. An abscission zone is formed between the fruit and rachis in *M. rotundifolia* but not in *Vitis* spp. The tendrils of *Vitis* spp. are branched but not in *M. rotundifolia*. From a genetic standpoint, *M. rotundifolia* has 20 chromosome pairs ($2n = 40$), similar to members of the *Ampelocissus* genus, whereas *Vitis* spp. possess 19 chromosome pairs ($2n = 38$). These differences have led multiple studies to classify *Muscadinia* as a separate genus, including Small (1913),

Bouquet (1978), and Olmo (1986). Based on these anatomical, biological, and genetic distinctions, we have followed the classification that recognizes *Muscadinia* as a distinct genus and have included these references in the manuscript to support this taxonomic distinction.

Page 3: “*Vitis* and its sister genus *Muscadinia* (Small, 1913; Bouquet, 1978; Olmo, 1986) diverged approximately 18-47 Mya (Wan et al., 2013; Liu et al., 2016; Ma et al., 2018) and each consist of dioecious species (Biasi and Conner, 2016)”.

(3) For the divergence of Viteae (*Vitis* and its allies), the authors may also check and cite Nie et al. Climate-influenced boretropical survival and rampant introgressions explain the thriving of New World grapes in the north temperate zone. *J Integr Plant Biol.*, 2023, 65(5): 1183-1203.

Thank you for the suggestion, the citation was added to the text.

(4) For SDR conservation with Vitaceae, I suggest the authors include the *Cissus rotundifolia* genome, which was published by Xin et al. 2022. The inclusion will expand the coverage of the phylogenetic diversity of *Cissus*, the largest genus of Vitaceae. Your current sampling includes two species both from the Neotropics. *Cissus rotundifolia* is from Africa, a region rich in *Cissus* (also the ancestral area of *Cissus*).

Thank you for the suggestion. The genome of *Cissus rotundifolia* (diploid species) published by Xin *et al.* (2022) was assembled as a haploid genome rather than a partially phased genome. Because the sequencing reads are not publicly available, we requested them to generate a partially phased assembly, but unfortunately, they were not provided by the authors. Consequently, we were not able to include this genome in our study.

(5) *Tetrastigma* is not really from tropical southern hemisphere areas (see page 15, line 417). *Tetrastigma* is widely distributed in subtropical and tropical Asia to Australia. So make revisions accordingly.

The sentence about the hypothesis that dioecy arose independently in the Vitaceae family based on their geographical locations was removed from the manuscript.

(6) P. 16, line457, change “Vitacea” to “Vitaceae”.

The correction was made.

Page 12: “However, DNA sequencing of additional samples from the hermaphrodite *Vitaceae* genera *Ampelocissus*, *Pterisanthes*, and *Nothocissus*, will be necessary to confirm it.”

(7) Supplementary Fig. 9, change “*V. sylvestris* ssp. *sylvestris*” to “*V. vinifera* ssp. *sylvestris*.” By the way, ssp. should not be italicized.

The correction was made in the caption of the supplementary figure: “Supplementary Fig. 11: Synonymous divergence between the F-associated and M-associated haplotypes in the sex-determining region in the males *V. arizonica* b40-14, *V. vinifera* ssp. *sylvestris* (*Vvs*) DVIT3351.27 and DVIT3603.16, and the *Muscadinia rotundifolia* (*Mrot*) Trayshed and Male DVIT1750.”

(8) Vitaceae in the title and in many parts of the manuscript should not be italicized.

Modification was made.

(9) Page 6, line 116, add “ancient” before “tracheophyte Selaginella...”.

The addition was made.

Page 4: “The dataset comprised 43 angiosperms (34 eudicots, one species of Ceratophyllales, four monocots, one species of magnoliids, one species of Chloranthales, two species from the ANA grade), and 13 non-flowering species: the gymnosperm *Taxus chinensis* (Chinese yew), the ancient tracheophyte *Selaginella moellendorffii* (spike moss), the bryophyte *Physcomitrium patens* (spreading earthmoss), and ten hornworts.”

(10) Page 6, line 116, change “embryophyte” to “bryophyte”. Embryophyte will be too broad for this context. You really mean bryophyte for *Physcomitrium*, at least it’s so much more precise systematically.

The modification was made, see (9).

(11) *Ampelopsis glandulosa* var. *brevipedunculata*, change to: *Ampelopsis glandulosa* var. *brevipedunculata*. The variety name *brevipedunculata* needs to be italicized. Also be sure this change is made in supplemental material, e.g., Supplemental Data 3.

The variety name was italicized in the manuscript, as well as Supplementary Data 3 and 6.

Page 13: “For long-read DNA sequencing, high molecular weight genomic DNA (gDNA) was isolated as in Chin *et al.* (2016) from *M. rotundifolia* Fry and DVIT1750, *A. aconitifolia* DVIT2492, *A. glandulosa* var. *brevipedunculata* PI 597579, *C. amazonica*, *C. gongylodes*, *P. quinquefolia* DVIT2400, *T. voinieranum*, and *L. coccinea* 1464”.

Page 13: “SMRTbell libraries of *M. rotundifolia* Fry and DVIT1750, *A. aconitifolia* DVIT2492, *A. glandulosa* var. *brevipedunculata* PI 597579, *C. amazonica*, *C. gongylodes*, and the HiFi libraries of *P. quinquefolia* DVIT2400, *T. voinieranum*, and *L. coccinea* 1464 were prepared as described in Minio *et al.* (2019; 2022), and sequenced using PacBio Sequel II system (Pacific Biosciences, CA, USA) at the DNA Technology Core Facility, University of California, Davis (Davis, CA, USA).”

(12) Change *Cayratia japonica* in Supplemental Data 3 -à *Causonis japonica*, to reflect the generic changes concerning *Cayratia*, see Wen *et al.* 2018, and Parmar *et al.* 2021 [Phylogeny, character evolution and taxonomic revision of *Causonis*, a segregate genus from *Cayratia* (Vitaceae), Taxon ..]. You actually used the correct names in Supplemental Data 6.

The name was modified in Supplementary Data 3.

Reviewer #3 (Remarks to the Author):

This manuscript presents a comparative analysis of the genetic sex-determining region in the grapevine *Vitis sylvestris* among Angiosperms. The first part of the manuscript consists of a broad comparative analysis of this region among angiosperms, showing that the region is conserved in flowering plants, but not in outgroups. The authors report that this is the first conserved region specialized in flowering that has been shown to be involved in sex determination. While the results on the conservation of this region are robust, it is unclear what this implies for sex chromosome

formation in plants, since all other known cases of sex-determining regions do not come from conserved regions.

To our knowledge, this is the first study to investigate the evolutionary conservation of a plant SDR across distantly related species. While previous studies have focused on individual SDRs, their broader conservation across angiosperms has not been systematically analyzed. Our findings reveal a case in which an established developmental network involved in flowering may have been repurposed for sex determination through modifications within this conserved locus.

However, we do not suggest that conservation of a genomic region is a universal prerequisite for SDR formation in plants. Whether the *Vitis* SDR represents an exception or part of a broader evolutionary pattern remains an open question. Future comparative analyses of additional plant SDRs across diverse taxa will be necessary to determine whether similar mechanisms underlie sex chromosome evolution in other plant lineages.

This point is discussed page 10: “The conservation of other plant SDRs and their role in flowering processes remain open areas for exploration. Evaluating the conservation of SDRs in distantly related plant species would help determine whether SDRs originate in highly conserved genomic regions across plants or if the *Vitis* SDR represents a unique case”.

In a second part, the authors study this sex-determining region in muscadine grapevines (*Vitis rotundifolia*) and in the genus *Tetragonia* of the family Vitaceae, both models having separate sexes. Based on genome assemblies, resequencing, polymorphism analysis and linkage disequilibrium, they find that the genus *Tetragonia* does not have the same sex-determining region that *V. sylvestris*. Of course, their data certainly allow a genome-wide approach that would allow the search for the sex-determining region in *Tetragonia*, but I imagine that will be done in a future study.

Concerning *Vitis rotundifolia*, the authors find that the same sex-determining region in *Vitis sylvestris* is most likely responsible for genetic sex. However, while I agree with their analyses, I disagree with their main conclusion, which is a unique origin of dioecy in the genus *Vitis*. They find that the 8-base pair deletion mutation present in INP1, the male candidate, is shared between the two *Vitis* species. This mutation therefore appeared prior to the separation between *V. sylvestris* and *V. rotundifolia*. However their own analyses contradict a single origin of dioecy in *V. sylvestris* and *V. rotundifolia*. Here is the list of arguments against this conclusion :

- they find that there is an inversion between the female haplotype and the male haplotype in *V. rotundifolia* while there is none in *V. sylvestris*. This suggests a different mechanism of recombination suppression in the two species.

There are two possible hypotheses regarding the role of the inversion between the F and M haplotypes in *Muscadinia rotundifolia*.

1. Inversion as the primary mechanism of recombination suppression: One possibility is that the inversion itself was a key event that initiated recombination suppression between the SDR haplotypes in *M. rotundifolia*. If this were the case, it would imply a different evolutionary trajectory for recombination suppression in *M. rotundifolia* compared to *Vitis* spp., which lacks such an inversion.

2. Inversion as a secondary event: Alternatively, the inversion may have arisen after the divergence of *Vitis* and *Muscadinia*, further reinforcing recombination suppression in *M. rotundifolia*. Under this scenario, an initial suppression mechanism, such as sequence divergence, epigenetic modifications, or structural variants other than an inversion, may have already been in place before the inversion occurred.

While the presence of an inversion in *M. rotundifolia* but not in *Vitis* spp. indicates that the two SDRs have undergone independent structural modifications, this does not necessarily contradict a shared evolutionary origin of dioecy. Instead, it suggests that different mechanisms may have contributed to the fine-scale regulation of recombination suppression after the divergence of these lineages. Further comparative studies, particularly those examining recombination rates and haplotype-specific expression patterns, will help clarify the role of the inversion in *M. rotundifolia* and its evolutionary significance for SDR function.

This point is discussed page 12: “Both inversions were found to have caused recombination suppression in the papaya SDR, suggesting that the inversion in the M haplotype of *M. rotundifolia* could play a similar role in the muscadine SDR. However, we estimated that the inversion occurred 40.3 ± 13.2 Mya, which overlaps with the divergence time between *Vitis* and *Muscadinia*. Another hypothesis is that the inversion in the M haplotype of *M. rotundifolia* occurred after the divergence between the two grape genera to further suppress recombination in the locus. This would explain why recombination completely stopped slightly earlier in *M. rotundifolia* (20.3 ± 8.0 Mya) compared to *Vitis* spp. (14.1 ± 8.2 Mya)”.

- they find that the age of the recombination suppression, infer from the synonymous polymorphism rate (dS) differs between *V. sylvestris* and *V. rotundifolia* (older in *V. rotundifolia*).

The synonymous polymorphism rate (dS) provides an estimate of when recombination completely ceased in a given region, rather than when recombination suppression initially began. Differences in dS between *Vitis* and *M. rotundifolia* suggest that the final stage of recombination suppression occurred earlier in *M. rotundifolia*, but this does not necessarily indicate independent origins of dioecy. Instead, it may reflect differences in the rate or extent of recombination suppression after the initial establishment of the SDR in their common ancestor.

Recombination suppression is often a gradual process involving multiple mechanisms accumulating over time. One key difference between these species is the presence of an inversion in the M haplotype of *M. rotundifolia*, which is absent in *Vitis* species. A plausible explanation for the difference in dS estimates is that this inversion arose after the divergence of the two genera, further reinforcing recombination suppression in *M. rotundifolia* but not in *Vitis* species. This hypothesis would explain why the estimated timing of full recombination loss appears older in *M. rotundifolia*.

This point is now explicitly discussed in the manuscript:

Page 12: “Another hypothesis is that the inversion in the M haplotype of *M. rotundifolia* occurred after the divergence between the two grape genera to further suppress recombination in the locus. This would explain why recombination completely stopped slightly earlier in *M. rotundifolia* (20.3 ± 8.0 Mya) compared to *Vitis* spp. (14.1 ± 8.2 Mya)”.

- they find that no gene except INP1 (responsible for male sterility) and the flaking gene Exostosin shows trans-specific polymorphism, which should be the case if the recombination suppression was prior to the separation.

In fact, their only argument for a common origin of dioecy (apart from the antiquity of male INP1) is the presence of sex-linked mutations in the promoter of YABBI3, the other candidate gene in the sex-determining region. But these results are only shown as a neighbor-joining tree, we would like to see the sequence alignment to see the raw results since it is so important.

I find that this study rather suggests that there were two independent recombination suppression events in *V. sylvestris* and *V. rotundifolia* that created these sex-determining regions, at the level of a region that had a genomic predisposition with a male-sterility mutation.

So I think the manuscript would be clearer if the authors reformulated their conclusions to be more in line with their own analyses, and showed more in-depth analysis of the YABBI3 promoter.

To address the reviewer's request for additional evidence on the *VviYABBY3* promoter, we have now included the sequence alignment of the 3-kb region upstream of the transcription start site in *Muscadinia rotundifolia*, *V. arizonica*, *V. vinifera* ssp. *sylvestris*, and *Vitis vinifera* ssp. *vinifera*. This alignment, which was used to generate the neighbor-joining tree, is now provided as Supplementary Fig. 15. In the figure, we highlight fifteen M-specific sites, which are marked with red squares. This new information has been incorporated into the manuscript.

Page 9: "Sequence alignment of the *VviYABBY3* promoter region revealed fifteen sites specific to the M alleles of *Vitis* and *M. rotundifolia* (Supplementary Fig. 15)".

We also added the coordinates of the M-specific MYB59-binding site in the sequence alignment, in which we can observe the impact of a M-specific SNP on TF-binding site.

Page X, line X: "A MYB59-binding site, located $2,104 \pm 14$ bp upstream of the TSS (from 2,366 to 2,373bp on the alignment (Supplementary Fig. 15)), was found specific to both *Vitis* and *M. rotundifolia* M alleles, *i.e.* present in all M haplotypes but absent in all F and H haplotypes".

Regarding the common origin of the *Vitis* and *M. rotundifolia* SDRs, we acknowledge that our previous interpretation of the synonymous divergence (*dS*) may have been unclear. To clarify this, we have rephrased the relevant sentence in the manuscript (page 8) as follows:

"In *Vitis* spp., the *dS* ranged from 0.0034 to 0.0543, while in *M. rotundifolia*, the range was higher, from 0.0174 to 0.064. These results suggest that recombination totally ceased slightly earlier in *M. rotundifolia* (20.3 ± 8.0 Mya) compared to *Vitis* spp. (13.8 ± 7.9 Mya)".

Additionally, we have now estimated the timing of the inversion in the M haplotype relative to the F haplotype. The inversion event is estimated to have occurred 40.3 ± 13.2 million years ago, a timeframe that overlaps with the estimated divergence time between *Vitis* and *Muscadinia* (18-47 Mya).

Page 8: "Based on the genetic distances between the F and M haplotypes at both ends of the inversion, we estimated that the structural variation occurred 40.3 ± 13.2 Mya".

Considering both the estimated timing of the inversion and the total cessation of recombination, we hypothesize that the inversion in the *M. rotundifolia* M haplotype arose after the divergence of the two grape genera. This suggests that while the initial SDR may have originated from a common ancestral state, lineage-specific structural changes, such as the inversion in *M. rotundifolia*, may have further reinforced recombination suppression independently in each lineage.

Page 12: “Another hypothesis is that the inversion in the M haplotype of *M. rotundifolia* occurred after the divergence between the two grape genera to further suppress recombination in the locus. This would explain why recombination completely stopped slightly earlier in *M. rotundifolia* (20.3 ± 8.0 Mya) compared to *Vitis* spp. (14.1 ± 8.2 Mya)”.

Authors' responses are in blue

Reviewer #1 (Remarks to the Author):

In the revised version of the manuscript "Evolutionary conservation of the grape sex-determining region in angiosperms and emergence of dioecy in Vitaceae", the authors have addressed several of my previous concerns, particularly regarding the novelty of this study compared to their 2020 publication. While their responses are generally satisfactory, I have a few minor suggestions for further improvement:

Conservation of the *Vitis* SDR and Flanking Regions: The authors state that the *Vitis* SDR consists of 12–14 protein-coding genes and used these genes to assess conservation across angiosperms. I recommend that the authors include a figure illustrating the collinearity results for these genes. Additionally, it would be valuable to investigate whether the genes flanking the SDR also exhibit collinearity and to what extent this conservation extends. Another important point is that the authors' evaluation of the functional significance of SDR conservation is not clear. The article found that grape SDR is highly conserved in angiosperms, but the significance and specific mechanism of this conservation in sex determination are difficult to explain. In the phylogenetic analysis, the authors expanded the gene set to 21, suggesting that the conservation might extend beyond the SDR. If so, the functional relevance of these flanking genes may not be limited to flower development, and this broader genomic context should be explored.

We agree that investigating broader conservation around the SDR is an interesting direction, however we believe it is beyond the scope of the present study and would distract from the primary objective. Our study is specifically focused on the conserved *Vitis* SDR itself, defined as the region from *VviYABBY3* to the *WRKY* transcription factor gene, comprising 12 protein-coding genes on haplotype 1 of Cabernet Sauvignon.

We have added a brief note in the discussion to acknowledge this as a potential area for future research.

“Investigating whether genes flanking the SDR also exhibit collinearity, and how far this conservation extends, could provide insight into whether additional gene functions beyond sex determination are conserved across flowering plants”.

In addition, to avoid any confusion or unintended interpretation beyond this defined region, we have removed any representation of flanking genes from the main figures (Figs. 2-4) and from the gene/repeat content shown for the *M. rotundifolia* SDR (Fig. 5). Phylogenetic and divergence analyses (Supplementary Figs. 11-15) are now limited to genes within the SDR or those directly relevant to the comparison between M and F haplotypes.

Estimation of Divergence Times: In the Results section, the authors inferred the timing of inversion events and haplotype differentiation based on ‘genetic distances’. However, they did not specify the exact genetic distances used, the sequences analyzed, or the reference points for calculating divergence times and their confidence intervals. This lack of detail could lead to misinterpretations regarding the origin of the SDR, the timing of recombination suppression, and the mutation rates

in coding regions and repetitive sequences. Clarifying these methodological details is crucial to ensure the robustness of the conclusions.

We have revised the manuscript to clarify the methodology used for estimating divergence times and to ensure transparency regarding the sequences and assumptions involved. Details about the identification of the inversion boundaries in the M haplotype relative to the F haplotype have been added to the Methods section. The coordinates of the inversion breakpoints in both haplotypes are now provided in Supplementary Table 2.

“To estimate the timing of the inversion in the M haplotype relative to the F haplotype in M. rotundifolia, we identified the boundaries of the inversion by aligning the M haplotype of Trayshed and DVIT1750 to their corresponding F haplotypes using NUCmer from MUMmer v4.0.0 with the “--mum” option (Marçais et al., 2018). Coordinates of the inversion ends are reported in Supplementary Table 2, and the associated sequences are provided in Supplementary Data 10”.

The sequences used for estimating the genetic distances, i.e. LTRs of retrotransposons and ends of the inversion, were added as Supplementary Data 9 and 10, respectively.

“To estimate the insertion time of LTR retrotransposons located between VviPLATZ1 and KASIII in the M haplotypes of M. rotundifolia, we first identified LTR elements using LTRharvest from the GenomeTools package v1.6.5 (Gremme et al., 2013). The paired LTR sequences for each retrotransposon (Supplementary Data 9) were aligned with MEGA X (Kumar et al., 2018) and genetic distances were calculated using Kimura’s two-parameter model (Kimura, 1980)”.

Analysis of *Tetrastigma* Species: I remain cautious about the analysis of *Tetrastigma* species. The authors performed linkage disequilibrium (LD) analysis by pooling data from 13 different species, but the distribution of SNP frequencies (which are likely species-specific) and the sequence divergence between species (which may not be related to sex linkage) could significantly bias the results. Therefore, the authors should exercise caution when interpreting these findings and avoid overgeneralizing the conclusions. It would be beneficial to provide a more detailed analysis of SNP frequencies and species-specific variations to validate the LD results.

To better assess potential biases, we analyzed the distribution of SNP positions identified in the 13 *Tetrastigma* species relative to the homologous region of the VSR in *T. voinianum*. We found that 61.2% of SNP positions were located within genes (Supplementary Fig. 6), and 27.2% of the SNPs were shared by at least two species. These findings, together with the limited genomic coverage of the samples (<3.9×), suggest that both low sequencing depth and interspecific sequence divergence likely constrained variant detection and may have introduced biases in LD estimation. Therefore, we have exercised caution in interpreting these results and now explicitly state in the revised text that conclusions from this analysis are preliminary and should not be overgeneralized. We also highlight the need for higher-quality, species-specific data to validate sex-linked signals in *Tetrastigma*.

“Within the VSR, 2,635 SNP positions were identified across the thirteen Tetrastigma species, with each species contributing an average of 326.6 ± 176.2 SNPs. The majority of SNPs (61.2%) were located within genes, and 27.2% were shared by at least two species (Supplementary Fig. 6).

However, only 63,490 bases, representing 9.23% of the VSR, were covered by at least one sequencing read in all thirteen species. This limited coverage likely reflects a combination of low genome coverage (< 3.9 X) and substantial sequence divergence across species, both of which may have constrained variant detection. The ratio of homozygous to heterozygous SNPs was significantly higher in Tetrastigma species compared to M. rotundifolia and Vitis spp. (Supplementary Fig. 7; Kruskal-Wallis test followed by post hoc Dunn's test; adjusted P < 0.05), indicating lower heterozygosity in Tetrastigma within the VSR. In established SDRs, elevated heterozygosity is typically observed due to the accumulation of sex-specific alleles, as recombination suppression limits genetic exchange between homologous chromosomes. The comparatively low heterozygosity observed in the Tetrastigma VSR suggests an absence of sex-specific divergence in this region, consistent with the lack of a differentiated SDR”.

Accordingly, we calculated the average r^2 per 1-kbp window having at least 5 SNPs. We found an average r^2 of 0.030 ± 0.081 per kbp, suggesting that the VSR in *T. voinieranum* is likely not under recombination suppression. We have moderated the interpretation of this result in the revised manuscript.

“To further investigate, we analyzed LD patterns in the VSR to evaluate potential recombination suppression, a hallmark of SDRs in dioecious species. Although the flower sex of T. voinieranum and the thirteen Tetrastigma species is unknown, strong LD would be expected in the SDR if recombination suppression were present, given that Tetrastigma is a dioecious genus. However, no such pattern was observed: the average r^2 was 0.005 ± 0.035 per kbp across the VSR (Fig. 4a; Supplementary Fig. 8), and 0.030 ± 0.081 per kbp when considering only 1-kbp windows with at least five SNPs. These findings suggest that the VSR in T. voinieranum is likely not subject to recombination suppression. Nonetheless, this conclusion should be interpreted with caution due to the limited sequence coverage and the availability of SNP data only across species”.

We added the following to the discussion to clarify the limitations of our approach:

“However, the genome coverage of the Tetrastigma species retrieved from NCBI was low (< 3.9X; Supplementary Data 6). Although we limited variant calling to species with at least 1.9 X coverage, only 63,490 bases out of 687.5 kbp were covered by at least one sequencing read in all thirteen species. This limited coverage, along with interspecific sequence divergence, likely constrained both variant calling and the LD analysis. To address these limitations, higher-quality, species-specific sequencing data from additional Tetrastigma species and populations will be necessary to confirm the absence of linkage constraints in this region. When combined with flower sex phenotyping, such data could facilitate the identification of the SDR in Tetrastigma”.

Additionally, we introduced language in the abstract to reflect this uncertainty:

“In the dioecious genus Tetrastigma, no evidence of recombination suppression was found in the homologous region, suggesting a potentially different mechanism of sex determination.”

The authors found 15 distant SNP sites on the promoter of VviYABBY3, one of which is a binding site for the MYB transcription factor, and speculated that its variation between M and F haplotypes may affect its expression level. The authors should verify the regulatory and activation capabilities

of this hypothesis, otherwise it is difficult to believe that the difference in expression of this gene between male and female organs is determined by the mutation of this site.

We agree that the proposed role of a MYB-binding site variant in the *VviYABBY3* promoter is speculative and should be interpreted with caution. We have revised the discussion to clarify that this hypothesis is based on *in silico* predictions and has not been experimentally validated. We now explicitly state that while this variation may contribute to differential expression between M and F haplotypes, functional assays are required to confirm its regulatory significance. The goal of including this observation is to highlight a possible mechanism that could be explored in future studies.

“Sequence differences between the M and F alleles of VviYABBY3 promoter region were found to affect the potential TF-binding sites in terms of TF and site numbers, suggesting a potential impact on the transcriptional regulation of the two alleles. In addition, the M allele of VviYABBY3 was detected as more highly expressed compared to its F counterpart in ovaries (Fig. 6b), raising the possibility that this allele may act as the female-suppressing gene in M. rotundifolia. However, no M-specific amino acid differences were found in the M allele of VviYABBY3 in Vitis and M. rotundifolia. Further investigation will be necessary to determine whether female suppression in the two grape genera is mediated by shared molecular mechanisms, and to clarify the transcriptional regulatory basis underlying the differential expression of the M and F alleles of VviYABBY3”.

Reviewer #4

[Editor: Reviewer #4 provides suggestion in Remark to Editor section. They think all previous suggestions from Reviewer #2 have been addressed.]

Reviewer #5 (Remarks to the Author):

Overall, the authors appear to have responded appropriately to the comments from Reviewer 3. However, one concern remains regarding the initial comment:

“While the results on the conservation of this region are robust, it is unclear what this implies for sex chromosome formation in plants, since all other known cases of sex-determining regions do not come from conserved regions.”

What Reviewer 3 is essentially asking here is what the relatively unchanged nature of the sex-determining region (SDR) in Vitaceae implies for the broader understanding of sex chromosome evolution in plants. This question arises because in most other dioecious plant species, SDRs tend to evolve rapidly, often undergoing large-scale degeneration (on the order of Mbp) and evolving independently. Consequently, the concept of conservation from an ancestral region is far from those cases. In this light, although the authors' manuscript demonstrates that the SDR in Vitaceae has been conserved from its ancestral state (and such case would be the first observed in the field

of plant sex determination), it does not clearly address what this finding means in the context of previously known patterns of sex chromosome evolution.

In the latter part of the authors' response, the authors refer to dS values, which is indeed relevant. However, the dS results suggest that, despite evolutionary time having passed, the changes in genes located in the SDR are minimal, and unlike in other plant species, a male-hemizygous state (if in a XY system) has not developed. This may represent a rare or exceptional case. Reviewer 3 may be implying that this unique feature of conservation could allow for a novel approach to sex chromosome analysis. Furthermore, there remain several misleading expressions in the revised manuscript. Some phrases could be misinterpreted to suggest that conservation exists between SDRs across different species, which likely reflects a conceptual gap between Reviewer 3 and the authors. I agree that the SDR in Vitaceae quite conserves the original genomic state, albeit not SDRs in other dioecious species. These should be revised for proper understanding. In reality, plant SDRs have evolved convergently from different genes and regions. If any "conservation" exists, it may only involve gene function (but note that, in the family Salicaceae or the genus Actinidia, SDRs undergo frequent turnover, which casts doubt on the conservation of even gene function across lineages).

Thank you for helping clarify Reviewer #3's comment. We now better appreciate the broader question regarding how the relative conservation of the *Vitis* SDR informs our understanding of sex chromosome evolution in plants, especially given that most known plant SDRs evolve rapidly and independently. We have revised the discussion accordingly:

“The high conservation and collinearity of the genes within the Vitis SDR locus among angiosperms suggest that this region has been under strong evolutionary pressure to maintain its structure and function (Fig. 2) (Tang et al., 2008). It would be interesting to investigate whether the genes flanking the SDR also exhibit collinearity, and to what extent this conservation extends. This would allow to explore whether other gene functions are conserved across plants. The higher rank-normalized gene conservation scores further imply that not only the arrangement but also the sequence of these genes is highly preserved, reinforcing the idea that this locus likely plays a critical role in flowering processes (Nair et al., 2022). The absence of a region collinear to the Vitis SDR in non-flowering plants further supports its potential specialized role in angiosperm reproductive development (Pellegrini et al., 1999). So far, the Vitis SDR is the only known plant SDR that exhibits high gene collinearity and conservation across angiosperms. In most plant lineages, SDRs evolve rapidly following recombination suppression, often resulting in large-scale degeneration of the M haplotype and hemizyosity for many genes (Ming et al., 2011). Consequently, gene content in the M haplotype is rarely conserved even among closely related species. To date, only the male-determining gene FrBy in kiwifruit has been shown to have orthologs in 32 angiosperm species, three of which retain similar function, suggesting limited functional conservation across lineages (Akagi et al., 2019). In contrast, the Vitis SDR, despite its estimated age (~20 Mya), shows minimal gene content divergence between F and M haplotypes. This unusual degree of conservation suggests that the Vitis SDR may contain genes essential for core developmental processes such as flowering. Functional studies, including targeted gene knockouts, will be critical to determine whether this constraint underlies the exceptional stability of the Vitis SDR. More broadly, investigating whether other plant SDRs originate from similarly conserved genomic regions or whether the Vitis SDR represents a unique

evolutionary case remains an open and compelling question in the study of plant sex chromosome evolution”.

In connection with Reviewer 3's comments, the following points should also be reconsidered:

Line 484: Use of the term "sexual dimorphism" (or "sexually antagonistic trait") is inappropriate. This term refers to traits that confer an advantage to only either of the two sexes, male or female (unrelated to sex determination itself), and there is no discussion of sexual dimorphism in this manuscript. The intended meaning here appears to be different from how the term is used.

We have replaced “dimorphism” with “dioecy” to more accurately reflect the intended meaning, which refers to the presence of separate male and female individuals rather than trait differences between sexes.

“To our knowledge, this is the first observation of dioecy emerging through the evolution of a genomic region which gene content is highly collinear and conserved in angiosperms and composed of multiple genes playing a role in flower development and morphology, and sexual fertility.”

Estimation of evolutionary timescales based on dS values: The interpretation presented in Line 663 seems to overestimate the timing of recombination suppression or species divergence. It may be possible that the nucleotide substitution rate was calculated "per generation" rather than "per year", which would significantly alter the conclusions regarding evolutionary timing.

Divergence times were calculated as $T = K/2\mu \times \text{generation time}$, where K is the genetic distance and μ is the mutation rate. We used a generation time of 3 years and a nucleotide substitution rate of 2.5×10^{-9} substitutions per base per year, as previously in Zou et al. (2021). In grapes, new seedlings produce flowers and fruit after 2-3 years. The point was clarified in the text.

“Divergence times were calculated as $T = K/2\mu \times \text{generation time}$, where K is the genetic distance and μ is the mutation rate. A generation time of 3 years and a nucleotide substitution rate of 2.5×10^{-9} substitutions per base per year were assumed (Zou et al., 2021)”.